# UpSkill: Mutual Information Skill Learning for Structured Response Diversity in LLMs

## Abstract

Reinforcement Learning with Verifiable Rewards (RLVR) has improved the reasoning abilities of large language models (LLMs) on mathematics and programming tasks, often by maximizing `pass@1` correctness. However, optimizing single-attempt accuracy can inadvertently suppress response diversity across repeated attempts, narrowing exploration and overlooking underrepresented strategies. We introduce UpSkill, a training time method that adapts *Mutual Information Skill Learning* (MISL) to LLMs to induce *structured response diversity*: a discrete latent $z$ selects a reproducible "strategy" that steers the token distribution toward distinct modes. We propose a novel reward that we implement within Group Relative Policy Optimization (GRPO): a *token-level* mutual information (MI) reward that encourages trajectory specificity to $z$. Experiments on GSM8K with three open-weight models, Llama 3.1–8B, Qwen 2.5–7B, and R1-Distilled–Qwen2.5–Math–1.5B show that UpSkill improves multi-attempt metrics for Qwen 2.5–7B, yielding gains of ∼4% in `pass@k` and ∼4% in `plurality@k` without degrading `pass@1`. Additionally, we prove that improvements in `pass@k` are closely tied to the mutual information objective, providing a theoretical justification for UpSkill.

## 1 Introduction

LLMs excel at verifiable reasoning tasks such as mathematical problem solving and code generation (Guo et al., 2025). However, repeated sampling often yields highly similar outputs (Shaier et al., 2025). This is detrimental in multi-attempt settings where just one correct completion solves the problem at hand, such as code generation with tests (Chen et al., 2021) or formal proofs in Lean (Trinh et al., 2024), as a lack of diversity reduces the effective number of independent attempts. Therefore, for these or other objectives evaluated by `pass@k`, or the probability that at least one of $k$ completions will be correct, more deterministic output decreases the chance that some sampled attempt will succeed. Furthermore, recent work has found that post-training that optimizes single-attempt correctness suppresses response variation across attempts (Chen et al., 2025; Dang et al., 2025), creating a discrepancy between how models are trained and how they are used and evaluated.

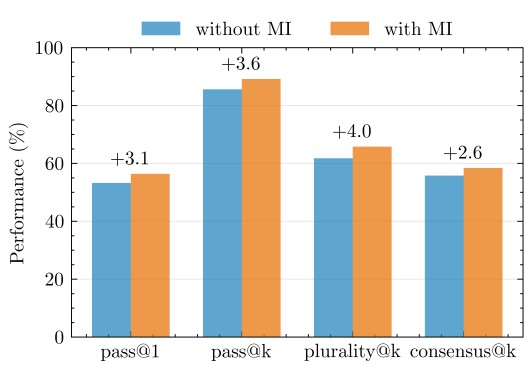

Figure 1: Token-level MISL improves multi-attempt accuracy without hurting single-attempt accuracy on GSM8K for the Qwen 2.5-7B model (See Sec. 5.2).

The challenge of balancing diversity and accuracy, or exploration and exploitation (Sutton and Barto, 2015), has primarily been studied in prior works that change how decoding is done. Methods such as temperature sampling (Renze and Guven, 2024), nucleus sampling (Holtzman et al., 2020), and prompt perturbations (Shur-Ofry et al., 2024) can inject variety, but they require manual tuning (Du et al., 2025) and are brittle across domains (Shi et al., 2024; Qiang et al., 2024). Separately, prior training-time methods do not expose a controllable basis of strategies and require a complex training

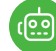

**Input**

If Raymond does half as much laundry as Sarah, and Sarah does 4 times as much laundry as David, calculate the difference in the amount of laundry Raymond and David do if Sarah does 400 pounds of laundry.

Sarah does 400 pounds of laundry.

David does 1/4 the laundry Sarah does, so he does 400 / 4 = 100 pounds of laundry.

Raymond does 1/2 the laundry Sarah does, so he does 400 / 2 = 100 pounds of laundry.

The difference is 200 − 100 = 100 pounds of laundry.

**Answer: 100 pounds**

Let D be the amount of laundry David does and let R be the amount of laundry Raymond does.

Since Sarah does 4 times as much laundry as David, and Raymond does half as much laundry as Sarah, we know that 4D = 400 and R = 400 / 2

The difference is:
R − D = 400 / 2 − 400 / 4 = 200 − 100 = 100

**Answer: 100 pounds**

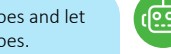

Figure 2: UpSkill is an unsupervised method for training LLMs to produce diverse responses. After training, different latent vectors $z$ (blue boxes above) correspond to different response strategies. Because of space constraints, the figure shows summarized responses from UpSkill; we report the full responses in Appendix K.

regime to properly balance exploration and exploitation (Tang et al., 2025; Chen et al., 2025). We seek a training-time mechanism that *(i)* increases diversity in a controlled manner, *(ii)* produces semantically distinct and reproducible modes of reasoning, and *(iii)* preserves single-attempt verifiable accuracy.

We introduce UpSkill, a training-time approach that induces *structured response diversity* without prompt engineering. The key idea behind UpSkill is to introduce an input token $z$ that structures the response, so that different values of $z$ correspond to different responses. Formally, we will model LLM attempts on verifiable reasoning tasks as a token-level Markov decision process. We can then adopt prior work from reinforcement learning on learning *skills*, which learn a policy conditioned on a latent variable $z$. These methods (Eysenbach et al., 2019; Gregor et al., 2016a; Achiam et al., 2018; Sharma et al., 2020a; Florensa et al., 2017) include a loss term that maximizes the mutual information between $z$ and the policy's behavior. Precisely, we adapt the CSF method (Zheng et al., 2024) to LLMs: the model conditions its response on a discrete latent $z \in \{1, \ldots, N\}$, and training encourages behaviors whose distribution depends strongly on $z$. Intuitively, each $z$ should correspond to a reproducible strategy, and the set of strategies should span a broad range of behaviors.

The main contribution of our paper is a method for training LLMs to produce diverse responses. Our method implements mutual information skill learning by applying GRPO (Shao et al., 2024) with a novel reward term: a token-level mutual information reward, which encourages diversity in completions. Finally, we sketch a theoretical link between $\mathcal{I}(\tau; z \mid x)$ and `pass@k`: the improvement of `pass@k` after training is related to $\mathcal{I}(\tau; z \mid x)$, In summary, our contributions are as follows:

- UpSkill achieves gains of +3.6% in `pass@k` and +4.0% in `plurality@k` on GSM8K on Qwen 2.5–7B using RL fine-tuning with LoRA adapters on 2,000 problems, with preserved `pass@1` accuracy. We additionally test two other open-weight models, although we do not observe improved performance.

- In an arithmetic puzzle environment, UpSkill improves `pass@5` by +10% by mitigating response variation collapse and developing a collection of diverse and complementary skills.

- We prove that `pass@k` improvement closely corresponds to the mutual information $\mathcal{I}(\tau; z \mid x)$, showing that large improvements in multi-attempt accuracy require — and are limited by — sufficient mutual information.

- We provide an effective and reproducible method for token-level MI, and will release an open-source implementation focused on practical performance.

## 2 BACKGROUND AND RELATED WORK

### 2.1 MULTI-ATTEMPT EVALUATION, REDUNDANCY, AND WHY DIVERSITY MATTERS

For verifiable tasks, we often consider the probability of success across *multiple* completions rather than a single attempt (Chen et al., 2025). Let $x$ denote the input and $\tau$ a sampled completion from policy $\pi(\cdot \mid x)$. Let $Y(\tau) \in \{0, 1\}$ indicate correctness under a deterministic verifier. For $k$ attempts, the standard metric

$$\text{pass@}k(x) \;=\; 1 - \Pr\left(\bigcap_{i=1}^{k}\{Y(\tau_i) = 0\}\,\bigg|\, x\right) \tag{1}$$

is the complement of the joint failure probability across $k$ i.i.d. draws $\tau_{1:k} \sim \pi(\cdot \mid x)$ (Chen et al., 2021). Letting $p = \Pr(Y(\tau) = 1 \mid x)$, we therefore have $\Pr(\text{pass@}k(x)) = 1 - (1 - p)^k$.

In practice, identical prompts with fixed decoding hyperparameters can yield strongly correlated trajectories, especially for deterministic or near-deterministic samplers (Vijayakumar et al., 2018). A useful lens is to consider an "effective number of attempts" $k_{\text{eff}}$ that discounts $k$ by a correlation term (analogous to design effects in sampling) (Kish, 1965). If completions have pairwise correlation $\rho$ in the binary success indicators, a heuristic adjustment gives $k_{\text{eff}} \approx k/(1 + (k-1)\rho)$: as $\rho \to 1$, additional attempts contribute little; as $\rho \to 0$, $k_{\text{eff}} \to k$. Although crude, this highlights the central point: reducing dependence among attempts is as important as raising per-attempt accuracy. Structured diversity aims to decrease redundancy so that the joint failure probability decreases faster in $k$. For Gaussian random variables, correlation and mutual information are closely related (as intuitively, correlated variables have information on each other) (Krafft, 2013). However, as text correctness cannot easily be framed as a Gaussian distribution, mutual information is more a natural measurement.

Beyond `pass@k`, `plurality@k` and `consensus@k` measure agreement among completions, examining robustness and internal consistency of the model's reasoning (Wallace et al., 2025). In many workflows, agreement acts as a proxy for confidence while still benefiting from diversity to escape shared failure modes (Hochlehnert et al., 2025).

### 2.2 RL ON LANGUAGE MODELS: TOKEN-LEVEL MDPS, RLVR, AND GRPO

Autoregressive LLMs can be cast as policies over a Markov decision process (MDP), where the state is the token prefix and the action is the next token (Bahdanau et al., 2017; Ouyang et al., 2022). This setup, often referred to as the token-level MDP (Zhong et al., 2025), allows reinforcement learning algorithms to directly optimize model behavior for correctness on verifiable tasks such as math or code.

Reinforcement Learning from Verifiable Rewards (RLVR) leverages automatically checkable signals (e.g., exact numeric answers, unit tests) as rewards, with the goal being to improve the pass rate of policies while ensuring the new policy remains close to a base model (Liu et al., 2023; Dou et al., 2024). Let $\pi_\theta$ denote the trainable policy and $\pi_{\text{base}}$ a frozen reference. A common form of the per-trajectory reward is

$$r_{\text{RLVR}}(\tau) \;=\; r_{\text{correctness}}(\tau) \;-\; \beta\, D_{\text{KL}}\big(\pi_\theta(\cdot \mid x) \,\big\|\, \pi_{\text{base}}(\cdot \mid x)\big), \tag{2}$$

where $\beta > 0$ controls deviation from the base model (Xiong et al., 2024).

Group Relative Policy Optimization (GRPO) (Shao et al., 2024) adapts PPO-style updates to reasoning by sampling multiple completions per prompt $x$ as a *group*. Within-group baselines reduce variance and increase the relative difference between completion rewards. Concretely, for each $x$ one draws $C$ trajectories $\{\tau_i\}_{i=1}^{C}$, computes verifiable rewards and a group baseline (e.g., a rank or mean-normalized signal), and updates $\pi_\theta$ with clipped policy ratios as in PPO (Schulman et al., 2017). GRPO typically improves `pass@1` on math/code under RLVR (Shao et al., 2024). However, absent any explicit term for diversity, it can *reduce* variation across attempts as the policy sharpens around locally high-reward regions (Dang et al., 2025).

As some intuition for this distribution change, suppose that the model is attempting to predict the correct answer in a setting where it believes that the answer is `Yes` with probability 70% and `No` with probability 30%. Cross-entropy loss encourages a model to predict the correct distribution of

70% `Yes` and 30% `No`; on the other hand, GRPO training would cause the model to collapse its output distribution towards predicting 100% `Yes`, as it maximizes the `pass@1`. Empirically, this can shrink the entropy of the completion distribution and heighten redundancy among attempts, limiting `pass@k` improvements even as `pass@1` increases (Dang et al., 2025).

## 2.3 MUTUAL INFORMATION AND SKILL DISCOVERY

Maximizing mutual information (MI) between latent variables and observed behavior has been a recurring tool for learning structured, controllable representations (Tishby et al., 2000; Kingma and Welling, 2013; Stratos and Wiseman, 2020).

In generative modeling, InfoGAN (Chen et al., 2016) augments GAN training with a variational lower bound on $\mathcal{I}(c; x)$ to make latent codes $c$ predictably control semantic factors (e.g., stroke thickness for MNIST). In variational autoencoders, InfoVAE (Zhao et al., 2018b) adds an explicit MI term to counteract posterior collapse and preserve informative latents even with expressive decoders.

In sequential decision making, MI has been used to discover diverse, reusable behaviors without external rewards. Early work such as VIC (Gregor et al., 2016b) and DIAYN (Eysenbach et al., 2018) maximizes $\mathcal{I}(s; z)$ or $\mathcal{I}(\tau; z)$, encouraging skills $z$ whose rollouts visit different parts of state or trajectory space and remain identifiable from observations. InfoGAIL (Li et al., 2017) extends this to imitation learning by maximizing MI between a latent intention and trajectories to capture multi-modal expert behavior. Subsequent methods bias the MI objective toward long-horizon distinctiveness to avoid trivial short-term variation (Sharma et al., 2020b; Hansen et al., 2021). Additional related work on MI is available in Appendix B.

Unsupervised skill discovery in RL can be viewed as maximizing the MI between a latent "skill" variable and observed trajectories (Gregor et al., 2016b; Eysenbach et al., 2018). Let $z \in \mathcal{Z}$ index a skill and let $\tau$ denote a trajectory. These methods maximize

$$\max_{\pi} \mathcal{I}(\tau; z \mid x) \;=\; \mathbb{E}\Big[\log p_{\pi}(\tau \mid x, z) - \log p_{\pi}(\tau \mid x)\Big] \;=\; \mathcal{H}(\tau \mid x) - \mathcal{H}(\tau \mid x, z). \tag{3}$$

This decomposition clarifies the pressure on the policy: *(i)* to increase marginal entropy $\mathcal{H}(\tau \mid x)$ so that trajectories cover more of the solution space; and *(ii)* to decrease conditional entropy $\mathcal{H}(\tau \mid x, z)$ so that each $z$ induces a reproducible, stable mode. The net effect is a set of distinct, consistent behaviors indexed by $z$ that together span diverse solution strategies.

Our setting is closest in spirit to unsupervised skill discovery (e.g., DIAYN, VIC) and to mutual information-based skill learning (Zheng et al., 2024), but differs in applying these techniques to language models with RLVR training. We develop an approach for maximizing MI tailored to LLM reasoning, and also connect `pass@k` performance with the mutual information objective.

## 2.4 OTHER TECHNIQUES FOR RESPONSE DIVERSIFICATION

Beyond MI-based training, several other approaches aim to increase output diversity.

At inference time, decoding-time diversification alters sampling: increasing temperature, switching to nucleus/top-$k$ sampling (Holtzman et al., 2020; Fan et al., 2018), or perturbing prompts (Qiang et al., 2024). While simple, these approaches face limitations: *(i)* they often fail to explore qualitatively distinct solution paths (Nguyen et al., 2025; Renze and Guven, 2024); *(ii)* they require domain-specific tuning (Wiher et al., 2022); and *(iii)* they can trade off against correctness and coherence (Nguyen et al., 2025). Prompt-cycling can inject domain knowledge (e.g., "try algebra" vs. "try geometry"), but it burdens users with prompt engineering and saturates well below human diversity (Shur-Ofry et al., 2024).

Determinantal point processes (DPPs) provide another path by rewarding sets of outputs that span high-volume regions in embedding space (Kulesza, 2012; Vijayakumar et al., 2018; Meister et al., 2023; Wang et al., 2024). In reinforcement learning, determinant-based rewards can encourage agents to explore trajectories that span complementary regions of state space (Ash et al., 2021; Zhao et al., 2024). Compared to MI, which directly couples a latent $z$ with trajectories to ensure reproducible modes, DPP-based diversity is distribution-free: it treats a set of samples as diverse if they occupy a high-volume region in representation space, regardless of whether the same diversity is reproducible under repeated sampling.

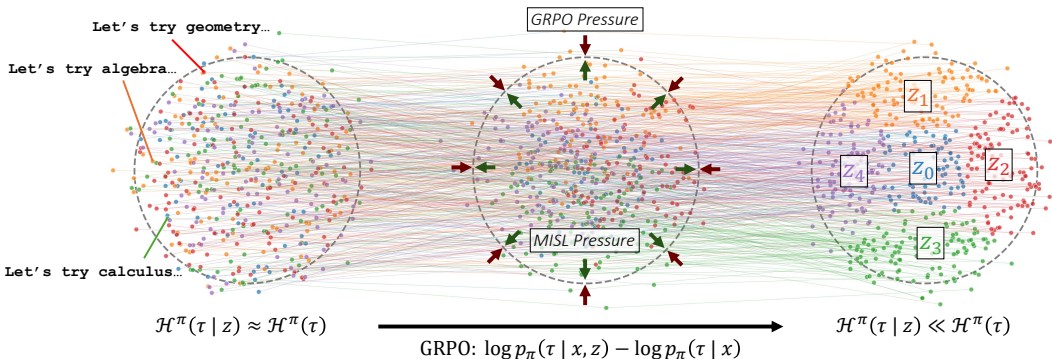

$$\mathcal{H}^\pi(\tau \mid z) \approx \mathcal{H}^\pi(\tau) \qquad\qquad \text{GRPO: } \log p_\pi(\tau \mid x, z) - \log p_\pi(\tau \mid x) \qquad\qquad \mathcal{H}^\pi(\tau \mid z) \ll \mathcal{H}^\pi(\tau)$$

Figure 3: Example illustration of how the MISL reward improves `pass@k` performance. Before MISL (left), the trajectory distribution is independent of the latents $z$, so the conditional entropy is close to the marginal. MISL training prevents distribution collapse due to `pass@1` training (middle). Adding the token-level MI reward (right) yields well-separated clusters indexed by $z$, reducing conditional entropy while preserving high marginal entropy. At inference, fixing different $z$ values produces consistent and diverse solution strategies.

Finally, training-time diversification has also been studied through explicit `pass@k`-based objectives. Tang et al. (2025) proposed an unbiased estimator for generic $k$-attempt objectives, showing overall improved model efficacy. Extending this, Chen et al. (2025) argue that simply training on `pass@1` falls victim to a local maximum of over-exploitation and reduced exploration. They find that `pass@k` training naturally focusing optimization efforts on harder problems producing significant improvements in both `pass@k` and `pass@1`. Outside of verifiable domains, DivPO (Lanchantin et al., 2025) alters preference optimization by contrasting diverse high-quality responses with common low-quality ones using a predefined diversity objective, yielding large diversity gains on creative and instruction-following tasks.

As we provide an orthogonal method to improve `pass@k` and diversity, our approach may complement that of Chen et al. (2025), Tang et al. (2025), and Lanchantin et al. (2025).

## 3 OPTIMIZING LLM DIVERSITY WITH MUTUAL INFORMATION

Given an input $x$ and an autoregressive policy $\pi(\,\cdot\,\mid x)$ that produces a completion (trajectory) $\tau = (y_1, \ldots, y_T)$, we introduce a *discrete* latent $z \in \{1, \ldots, N\}$ via a lightweight prompt prefix (e.g., `Strategy {z} |`), yielding conditional policies $\pi(\cdot \mid x, z)$. During training, $z$ is drawn uniformly at random from the set $\{1, \ldots, N\}$. At inference, one selects $k \le N$ distinct values of $z$ and generates one completion per value, producing $k$ semantically distinct attempts.

### 3.1 OBJECTIVE

We would like to encourage *structured response diversity* by maximizing the conditional mutual information $\mathcal{I}(\tau; z \mid x)$. Intuitively, maximizing mutual information makes the outputs of different strategies distinguishable, ensuring that each $z$ induces a reliably different mode. By querying each strategy once, we obtain $k$ semantically distinct attempts. Formally, this corresponds to maximizing

$$\max_\pi \ \mathcal{I}(\tau; z \mid x) \ = \ \mathbb{E}\Big[ \log p_\pi(\tau \mid x, z) - \log p_\pi(\tau \mid x) \Big], \tag{4}$$

which increases the overall entropy of trajectories while reducing the conditional entropy within each $z$-mode, ensuring diverse yet reproducible strategies. The term $p_\pi(\tau \mid x) = \frac{1}{N}\sum_{z'=1}^{N} p_\pi(\tau \mid x, z')$ is a uniform mixture over skills. Maximizing mutual information encourages *(i)* high marginal entropy of trajectories, promoting broad coverage; and *(ii)* low conditional entropy given $z$, so that each response is distinct and determined by $z$. Figure 3 provides an overview of the relevant dynamics.

---

**Algorithm 1** UpSkill: A method for training LLMs to produce diverse responses with mutual information.

---

1: **Inputs:** base policy $\pi_{\text{base}}$, trainable policy $\pi$, latent count $N$, completions per group $C$, weights $(\alpha_1, \alpha_2, \beta)$
2: **repeat**
3:    Sample a minibatch of prompts $\{x\}$
4:    **for** each $x$ in the minibatch **do**
5:       Sample $z \sim \text{Unif}(\{1, \ldots, N\})$; generate $C$ completions $\{\tau_i\}_{i=1}^C$ with $\pi(\cdot \mid x, z)$
6:       Compute $r_{\text{corr}}(\tau_i)$, $r_{\text{TMI}}(\tau_i; x, z)$, and $\Delta_{\text{KL}}(\tau_i)$ as above
7:    **end for**
8:    Form per-sample rewards via equation 6; compute advantages; update $\pi$ with GRPO
9: **until** convergence

---

## 3.2 Token-level mutual information reward

We now focus on implementing the mutual information as a token-level reward. For each pair $(x, z)$, let $\{\tau_i\}_{i=1}^C$ be $C$ completions sampled from $\pi(\cdot \mid x, z)$. We define a *per-sample* token-level score

$$r_{\text{TMI}}(\tau_i; x, z) \;=\; \sum_{t=1}^{C} \Big[ \log p_\pi(y_t \mid x, z, y_{<t}) - \log p_\pi(y_t \mid x, y_{<t}) \Big], \qquad (5)$$

where the second term is the uniform mixture

$$p_\pi(y_t \mid x, y_{<t}) \;=\; \frac{1}{N} \sum_{z'=1}^{N} p_\pi(y_t \mid x, z', y_{<t}).$$

Log-probabilities are computed by $\pi$ on the realized $\tau_i$. In our implementation the mixture is computed *exactly* across all $N$ skills; this is feasible for the $N$ used in our experiments (Section 5). Since $\frac{1}{C} r_{\text{TMI}}(\tau_i; x, z)$ is a Monte Carlo estimator of $\mathcal{I}(\tau; z \mid x)$, we make this our main reward term with UpSkill, with the other reward term being considered in ablation experiments. Appendix D discusses an alternative based on semantic mutual information.

## 3.3 Combined RL objective

Let $r_{\text{corr}}(\tau_i) \in \mathbb{R}$ denote the verifiable correctness reward (often binary) and define the per-trajectory KL penalty as

$$\Delta_{\text{KL}}(\tau_i) \;=\; \sum_{t=1}^{|\tau_i|} \log \frac{\pi(y_t \mid x, z, y_{<t})}{\pi_{\text{base}}(y_t \mid x, z, y_{<t})}.$$

The per-sample scalar reward is

$$r(\tau_i; x, z) \;=\; r_{\text{corr}}(\tau_i) \;-\; \beta \, \Delta_{\text{KL}}(\tau_i) \;+\; \alpha_1 \, r_{\text{TMI}}(\tau_i; x, z), \qquad (6)$$

with $\alpha_1, \beta \geq 0$. We apply GRPO to optimize the sum of the combined rewards.

## 3.4 Training procedure

We fine-tune a trainable policy $\pi_\theta$ with GRPO while injecting a discrete strategy variable $z \in \{1, \ldots, N\}$. At each step, we draw a minibatch of prompts $x$ and, for each $x$, sample a strategy $z$ uniformly and generate $C$ completions $\tau_{1:C} \sim \pi_\theta(\cdot \mid x, z)$ under fixed decoding. For every completion $\tau$, we compute: *(i)* a verifiable correctness reward $r_{\text{corr}}(\tau)$ from the task's deterministic checker; *(ii)* the token-level MISL term $r_{\text{TMI}}$ that measures how specific the trajectory is to the chosen strategy; and *(iii)* a KL control term toward a frozen base policy. We then update the policy with GRPO on this reward.

## 3.5 Inference

Given a budget of $k$ attempts, we choose $k$ distinct latents from $\{1, \ldots, N\}$ and generate one completion per latent under fixed decoding hyperparameters. Aggregation (e.g., majority vote) can optionally be applied. Because each completion is produced by a trained, distinct mode, conditional success probabilities remain larger than with redundant samplings, improving multi-attempt metrics.

## 4 THEORETICAL CONNECTION BETWEEN PASS@K IMPROVEMENT AND MUTUAL INFORMATION

Our main theoretical result shows that the mutual information objective is closely tied to `pass@k`. In particular, we will show that the mutual information objective is a lower bound on *improvement* in the `pass@k` objective, so maximizing mutual information provably results in an increased (lower bound on) `pass@k`. Our theoretical results will require the following assumptions:

1. $k$-uniform mixture model: Assume that the marginal distribution over the skills is identical to the base model.

2. Distributional impact: Let $a_z$ be the probability of success of strategy $z$ and $a$ be the probability of success of the base model. Assume that for all $x \in \mathcal{X}$ there exists $\eta > 0$ such that for all $z \in [k]$, $|a_z - a| \geq \eta \delta(\pi_{M,z}(\cdot \mid x), \pi_B(\cdot \mid x))$, where $\delta$ is the total variation distance.

The second assumption says that the distribution shifts induced by UpSkill correspond to different problem approaches, and, as a result, will have different probabilities of success. A more precise definition of $k$-uniform mixture models, additional justification for the assumptions, and the statement and proof of the lemma are in Appendix C.

**Lemma 1.** Let `pass@k`$_B$ be the `pass@k` score of the base model on prompt $x$ and `pass@k`$_M$ be the `pass@k` score of the mixture model on prompt $x$. Under the above assumptions, we show that:

$$1 - \exp\left(-C_1 \eta^2 \mathcal{I}(\tau; z \mid x)^2\right) \leq \frac{\texttt{pass@k}_M - \texttt{pass@k}_B}{1 - \texttt{pass@k}_B} \leq 1 - \exp\left(-C_2 \mathcal{I}(\tau; z \mid x)\right)$$

where $C_1$ depends on $k$ and $C_2$ depends on $k$ and $\max_z a_z$.

The quantity in the middle can be interpreted as the fraction of possible improvement from the base model that is realized by the mixture model. Since monotonically increasing functions in $\mathcal{I}(\tau; z \mid x)$ provide both lower and upper bounds on how much the mixture model improves over the base model, it makes sense to optimize directly for the mutual information. UpSkill explicitly increases $\mathcal{I}(\tau; z \mid x)$ during training, ensuring diversity across skills and giving a guaranteed improvement in `pass@k` over the base model.

## 5 EXPERIMENTS

We evaluate whether conditioning on a discrete latent $z$ and training with our token-level mutual-information reward equation 5 improves multi-attempt metrics. We present results in two settings: (1) a controlled arithmetic environment that allows for fully verifiable evaluation and direct inspection of distributional effects, and (2) the GSM8K benchmark across three open-weight models. We then report ablations on the number of strategies $N$ and on adding a semantic-MI surrogate.

### 5.1 ARITHMETIC ENVIRONMENT

The arithmetic environment consists of prompts with three single-digit integers and a latent skill index $z \in \{0, \ldots, N-1\}$. A small transformer model chooses one of the integers to be a target and is required to produce a simple arithmetic expression with the other two digits that evaluates to this target. We use an automatically verified correctness reward, and training uses GRPO without a KL penalty. Full details of the environment, model, and training procedure are provided in Appendix E.

Figure 4 illustrates that in the control condition ($\alpha_1 = 0$), training quickly collapses to a single deterministic strategy: by the end of training, `pass@1` and `pass@5` are indistinguishable (0.793), offering no benefit from multiple attempts. In contrast, with UpSkill ($\alpha_1 = 0.5$, MI reward cap at 1.0), the model maintains diverse trajectories, yielding a substantially higher multi-attempt accuracy (`pass@5` = 0.897) despite a lower single-attempt accuracy (`pass@1` = 0.390). This difference aligns with the entropy dynamics: UpSkill preserves broad output distributions (token entropy changed from 0.723 to 0.797 over training), sustaining diverse strategies and higher `pass@5`. By contrast, the control run—while substantially improving `pass@1`—collapses to near-deterministic

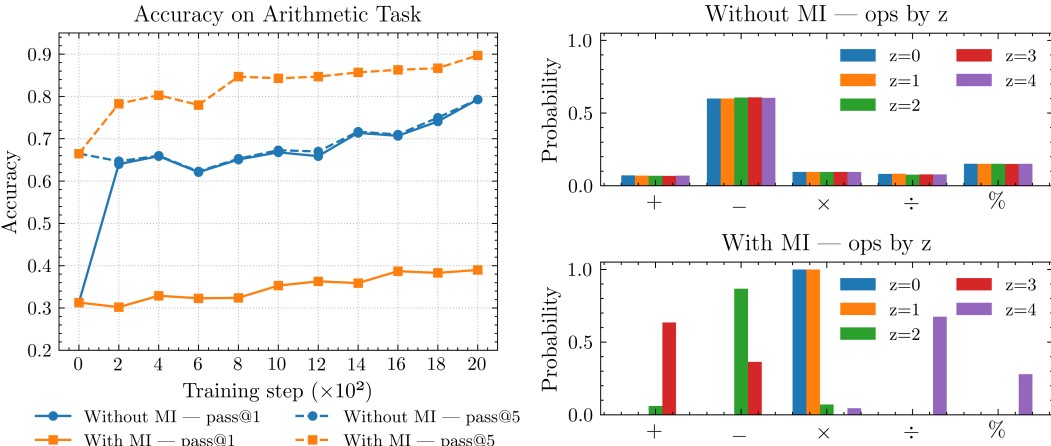

Figure 4: Arithmetic environment results. Training curves show that under GRPO alone (blue), `pass@1` and `pass@5` converge together, indicating that multiple attempts provide little benefit. With MISL (orange; $N$=5), `pass@5` improves substantially while `pass@1` remains modest, demonstrating that different latents yield complementary solutions. Operator distributions further highlight this effect: without MISL, they are nearly identical across $z$, reflecting a lack of specialization, whereas with MISL, distinct latents focus on different operators, producing diverse strategies that drive multi-attempt gains.

outputs (with entropy 0.723 changing to 0.030), leaving `pass@5` identical to `pass@1` (see Appendix E.5 for more details).

Figure 4 illustrates that under UpSkill, different $z$ values yield distinct distributions over operators and digits, whereas the control produces nearly identical distributions across $z$. In this small-scale environment, we can directly observe the learned strategies, and notably $z = 1$ and $z = 2$ converge to risky yet common modes, whereas other values of $z$ cover the remaining operations, improving multi-attempt success with a strategy infeasible for optimizing a `pass@1` objective. The distributions over the first digit are available in Appendix E.7.

We additionally ablate the impact of starting model capabilities, the coefficient of KL penalty, and GRPO parameters (see Appendix F for full details and results). KL penalty $\beta$ discourages entropy collapse by ensuring the new policy remains close to the initial policy, thereby improving performance. On models with $\beta \in [0.05, 0.10]$, we test $\alpha_1 \in [0.1, 0.3, 0.5]$ and find that well-chosen MI-reward parameters increase `pass@k` by an average of 3% for the weaker base model. However, for the stronger base model, we find the opposite trend. It is always best to choose $\alpha_1 = 0$, with the best choice of $\alpha_1 \in [0.1, 0.3, 0.5]$ still leading to a 1.2% performance decrease. Our theoretical results in Lemma 1 suggest that UpSkill improvement is negatively related to `pass@1` (equivalently `pass@k`$_B$) capability, and thus, although surprising, this result is in line with our theoretical analysis. We separately conjecture that $\beta$, which corresponds with a decrease in exploration from the base policy, conflicts with the mutual information incentive to explore.

## 5.2 GSM8K

We next evaluate on GSM8K (Cobbe et al., 2021), a dataset of grade-school arithmetic word problems. We use 2,000 training problems and a held-out set of 500 questions. All experiments are conducted in a zero-shot setting with a maximum sequence length of 1024 tokens. We train LoRA adapters (approximately 80M trainable parameters) on top of three open-weight backbones: Llama 3.1–8B (Grattafiori et al., 2024), Qwen 2.5–7B (Qwen et al., 2025), and R1-Distilled–Qwen2.5–Math–1.5B (DeepSeek-AI et al., 2025). As before, we apply GRPO with a correctness reward and with default KL penalty, and UpSkill is applied at the token level as in Eq. (6). At inference, we fix $k$ distinct skill indices and generate one completion per skill. More details and ablations are available in Appendices G, H, and I. Figure 5 shows our performance on the withheld evaluation set. Token-level MISL improves results on Qwen 2.5–7B on all metrics. The other tested models are generally

unaffected. Interestingly, these results do not entirely parallel the arithmetic environment, as UpSkill's improvement on Qwen 2.5–7B has not come at the cost of `pass@1`. We hypothesize this is due to the existence of a larger set of correct reasoning approaches, and thus MISL does not necessarily come at a cost of `pass@1`. Chen et al. (2025) have separately found that `pass@k` training methods can improve `pass@1` performance. We include summarized outputs from Qwen in Figure 2 and the isolated Qwen results in Figure 1.

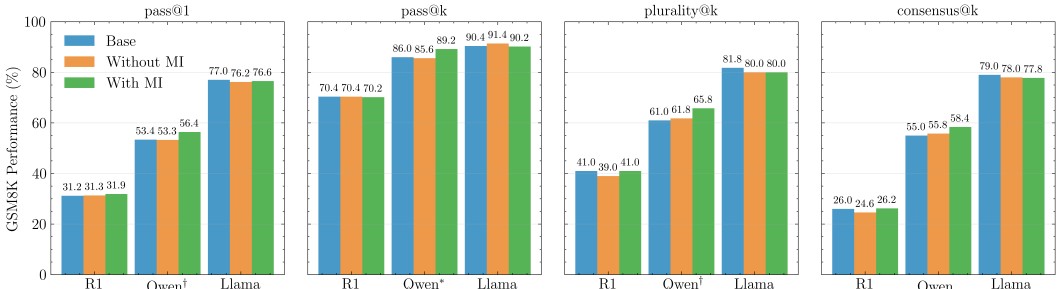

Figure 5: Performance on $500$ held-out problems with $N{=}5$ strategies. We observe gains on all metrics for the Qwen model. `Base` refers to the model before GRPO training, `Without MI` refers to after GRPO training without token MI, and `With MI` refers to training with GRPO and token MI. An asterisk denotes $p<0.02$ and † denotes $p<0.06$ from McNemar's test on the hypothesis 'With MI' outperforms 'Without MI'.

### 5.3 GSM8K ABLATIONS

We conduct two ablations to probe the robustness of the approach. First, increasing the number of skills beyond $N{=}5$ produces mixed results. Many GSM8K problems admit only a limited number of distinct solution paths, so larger $N$ values fragment the capacity into modes that do not translate into additional gains. Second, we study the effect of replacing the token-level MI with a semantic MI, which we formally introduce in Appendix D. This semantic MI occasionally yields further improvements but introduces instability due to estimator variance in high dimensions. We provide more details in Appendix J. Therefore, our main results use token-level MISL only, with semantic variants left as a direction for future work.

## 6 CONCLUSION

Our experiments show that UpSkill provides a simple and effective way to induce strategy-level diversity in LLMs, leading to gains on multi-attempt metrics such as `pass@5` and `plurality@5`. By conditioning on discrete latent variables, the model learns reproducible modes of reasoning that reduce redundancy across attempts and increase the likelihood of success. Beyond these empirical findings, UpSkill provides a principled training-time approach for improving response diversity, and our analysis links $\mathcal{I}(\tau; z \mid x)$ to upper bounds in `pass@k` improvement in training. We hope this stimulates research on robust semantic diversity signals and theoretical ties between information-theoretic objectives and multi-attempt success.

**Limitations.** There are a few notable limitations with our work. Assumption 1 is rather limiting and difficult to enforce empirically, so we hope to find a natural way to either incorporate it into our training method or find another assumption that can also prove Lemma 1. As future work, we hope for further experimentation across domains and with larger models, to better study how our method is affected by KL penalty and base model performance, and to approach the theoretical guarantees from the perspective of Tsallis entropy (Furuichi, 2006). The Llama and R1 models had unstable RL training, causing performance to degrade from the base model; we will switch to a more reliable RL framework like verl (Sheng et al., 2024) and use a more robust prompting method than an already semantically meaningful prefix string.

**Reproducibility.** We make several efforts towards encouraging reproducibility of our work and results. **We have included all experiment code in the supplementary files**, along with instructions to reproduce our results. We use only models under permissive licenses. For our theoretical results, we provide a description of assumptions and the complete proof of claims in the appendix. Additionally, for various experiments whose full results could not fit in the main paper, we include the full results in the appendix.

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

## A    USE OF LLMs

Large language models were used in the preparation of this work for writing assistance (including polishing, improving presentation of concepts, and restructuring of text), for retrieval and discovery of related work, and for support in producing experimental code and figures. Language models were additionally used for feedback on the paper and to formalize mathematical arguments. All analysis, experimental and method design, and final interpretations are our own. LLM outputs were always rigorously reviewed.

## B    EXTENDED MUTUAL INFORMATION RELATED WORK

Estimating MI reliably is challenging in high dimensions. Variational bounds (Barber–Agakov) optimize a classifier or regressor $q_\phi(z\,|\,\cdot)$ as a proxy for the intractable posterior (van den Oord et al., 2019). Contrastive bounds such as InfoNCE (van den Oord et al., 2019) reduce MI estimation to noise-contrastive classification and have become standard due to their stability. Neural MI estimators like MINE (Belghazi et al., 2021) directly optimize a Donsker–Varadhan bound but can suffer from bias/variance trade-offs and training instability. Nonparametric $k$NN estimators (KSG) (Kraskov et al., 2004) avoid parametric critics but require many samples and are sensitive to dimension, motivating careful batching and normalization when used inside policy gradients. In text generation, MI-style objectives have been used to prevent latent collapse and enable controllable generation, e.g., by encouraging informative latents in variational text models (Zhao et al., 2017; 2018a) or aligning codes with style attributes (John et al., 2018). These approaches typically maximize MI between prompts or attributes and latent variables, rather than between a discrete strategy and the full trajectory distribution, and are optimized with supervised losses rather than RL.

Conceptually, our objective reconciles two desiderata emphasized in prior MI work: *coverage* (high marginal entropy over trajectories) and *control* (low conditional entropy given $z$). Whereas decoding-time diversity manipulates token entropy without guarantees about identifiable modes, MI-based diversification learns *reusable, reproducible* modes indexed by a small discrete latent. This makes diversity a first-class, training-time property that can be cleanly exercised at inference by selecting distinct $z$ values.

## C    STATEMENT AND DERIVATION OF THEORETICAL BOUNDS

### C.1    PROBLEM SETUP AND ASSUMPTIONS

Let $\mathcal{X}$ be the set of all possible prompts. The statement of Lemma 1 applies to the general class of *$k$-uniform mixture models*.

**Definition.** A $k$-uniform mixture model $(M, B)$ is defined to be an ordered pair of a *mixture model*, which is a set of $k$ different policies for generating trajectories, which we will call $\pi_{M,z}(\cdot \mid x)$ for a prompt $x \in \mathcal{X}$ and strategy $z \in [k]$, along with a *base model* $\pi_B(\cdot \mid x)$ for generating trajectories subject to the condition that

$$\frac{1}{k} \sum_{z=1}^{k} \pi_{M,z}(\cdot \mid x) = \pi_B(\cdot \mid x) \, \forall \, x \in \mathcal{X}.$$

This definition can be interpreted as follows: $\pi_B$ is the trajectory distribution of the original, non-strategy conditioned language model. If we weigh each strategy as being equally important, we sample once from the mixture model by randomly choosing one strategy. In this case, the joint distribution of trajectories from the mixture is

$$\pi_M(\cdot \mid x) := \frac{1}{k} \sum_{z=1}^{k} \pi_{M,z}(\cdot \mid x).$$

The condition essentially means that the joint distribution of trajectories over picking a strategy uniformly at random must be the same as the original distribution. Therefore, in essence, the mixture model partitions $\pi_B$ into $k$ different policies that together average back to $\pi_B$.

In practice, this condition imposes undue constraints on the strategy distribution, and thus, for the practical implementation, this is not enforced. Also, while in practice one may actually train $N > k$ different strategies and then randomly sample $k$ different strategies so that they still have equal probabilities of being selected, here we make the simplifying assumption that $N = k$, which is true for all of our experiments.

For ease of notation, let $a = \Pr(Y_x(\tau_{B,1}) = 1 \mid x)$ and $a_z = \Pr(Y_x(\tau_{M,z}) = 1)$ for $z \in [k]$. Because the trajectories $\tau_{M,z}$ are sampled independently, we have that

$$\pi_M(\cdot \mid x) = \pi_B(\cdot \mid x) \implies \frac{1}{k} \sum_{z=1}^{k} a_z = a. \tag{7}$$

We provide additional justification for the second assumption made in section 4. First, if the strategies differ in more than just style and contain meaningful semantic differences, we expect that the difference in success should be proportional to how different these two distributions are. The total variation distance measures this distance in trajectory space, while the constant $\eta$ controls how sensitive the success probabilities are to changes in the distribution shift.

## C.2 EXTENDING `PASS@K` TO $k$-UNIFORM MIXTURE MODELS

We now extend the traditional definition of `pass@k` to fit the setting of $k$-uniform mixture models to leverage the fact that we now have a natural structure for querying $k$ different strategies by varying $z$. This is notably different from the setting in the consistency assumption, which can be interpreted as querying just one strategy uniformly at random.

For a given prompt $x$ and deterministic verifier $Y_x(\tau)$ outputting 1 if $\tau$ is a valid output on prompt $x$ and 0 otherwise, and $k$-uniform mixture model $(M, B)$, define `pass@k`$_M$ be the probability that querying each of these strategies independently exactly once (see Sec. 3.5) results in at least one correct answer, and define `pass@k`$_B$ to be the probability that querying $\pi_B$ independently $k$ times results in at least one correct answer. Writing this out mathematically, for each $z \in [k]$ we sample $\tau_{M,z} \sim \pi_{M,z}(\cdot \mid x)$; then

$$\texttt{pass@k}_M \ = \ 1 - \Pr\left(\bigcap_{z=1}^{k} \{Y_x(\tau_{M,z}) = 0\} \, \middle| \, x\right) = 1 - \prod_{z=1}^{k} \Pr(Y_x(\tau_{M,z}) = 0 \mid x). \tag{8}$$

While we use the same definition of `pass@k` for $B$ as in standard literature, we include it for the sake of completeness; similarly, when sampling $\tau_{B,z} \sim \pi_B(\cdot \mid x)$ independently for each $z \in [k]$, we find that

$$\texttt{pass@k}_B \ = \ 1 - \Pr\left(\bigcap_{z=1}^{k} \{Y_x(\tau_{B,z}) = 0\} \, \middle| \, x\right) = 1 - \Pr(Y_x(\tau_{B,1}) = 0 \mid x)^k \tag{9}$$

where we simplify the product into the RHS of equation 9 with the independence of samples. Then

$$\texttt{pass@k}_B = 1 - (1-a)^k, \quad \texttt{pass@k}_M = 1 - \prod_{z=1}^{k} (1 - a_z).$$

We now give the precise statement of the main result.

**Lemma 2** (`pass@k` Improvement for $k$-uniform Mixture Models, Full Statement). Let $u = \max_{z \in [k]} a_z$ and let $\varphi : \mathbb{R}_{\geq 0} \mapsto \mathbb{R}$ be defined as $\varphi(x) = \frac{x \log x}{x-1}$ for $x \neq 0, 1$ and $\varphi(0) = 0, \varphi(1) = 1$. Then

$$1 - \exp\left(-\frac{k\eta^2 \mathcal{I}(\tau; z \mid x)^2}{2\varphi(k)^2}\right) \leq \frac{\texttt{pass@k}_M - \texttt{pass@k}_B}{1 - \texttt{pass@k}_B} \leq 1 - \exp\left(-\frac{k\mathcal{I}(\tau; z \mid x)}{4(1-u)^2}\right).$$

## C.3  Derivation of Lower Bound

Using Taylor's Theorem on $f(y) = \log(1-y)$ gives the equations $f(a_z) = f(a) + (a_z - a)f'(a) + \frac{1}{2}(a_z - a)^2 f''(\xi_z)$ where $\xi_z$ lies in between $a_z$ and $a$, for $z \in [k]$. Summing all of these equations, the linear terms cancel due to equation 7. Then

$$\sum_{z \in [k]} \log(1 - a_z) = k \log(1-a) + \sum_{z \in [k]} \frac{1}{2}(a_z - a)^2 f''(\xi_z). \tag{10}$$

Let $A$ be the random variable that takes value $a_z$ for each $z \in [k]$ with probability $\frac{1}{k}$. Since $f''(y) = -\frac{1}{(1-y)^2}$ is a decreasing function in the interval $[0, 1)$, we find that $f''(\xi_z) \leq f''(0) = -1$ for all $z \in [k]$, i.e.

$$\sum_{z \in [k]} \log(1 - a_z) \leq k \log(1-a) - \sum_{z \in [k]} \frac{1}{2}(a_z - a)^2$$

$$\implies k \log(1-a) - \sum_{z \in [k]} \log(1 - a_z) \geq \frac{k\text{Var}(A)}{2} \implies \frac{1 - \texttt{pass@k}_B}{1 - \texttt{pass@k}_M} \geq \exp\left(\frac{k}{2}\text{Var}(A)\right)$$

$$\implies \texttt{pass@k}_M - \texttt{pass@k}_B = (1 - \texttt{pass@k}_B)\left(1 - \frac{1 - \texttt{pass@k}_M}{1 - \texttt{pass@k}_B}\right)$$

$$\geq (1 - \texttt{pass@k}_B)\left(1 - \exp\left(-\frac{k\text{Var}(A)}{2}\right)\right).$$

We now find a lower bound for $\text{Var}(A)$ in terms of the mutual information to finish off the proof. Let $\delta(\cdot, \cdot)$ represent the total variation distance between two distributions. We have that $\text{Var}(A) \geq \mathbb{E}[|A|]^2$ by Jensen's Inequality, so using the distributional impact assumption,

$$\text{Var}(A) \geq \left(\mathbb{E}_{z \sim \text{Unif}\{1,2,\ldots,k\}}[|a_z - a|]\right)^2 = \left(\frac{1}{k}\sum_{z=1}^{k}|a_z - a|\right)^2 \geq \left(\frac{1}{k}\sum_{z=1}^{k}\eta\delta(\pi_{M,z}, \pi_B)\right)^2$$

$$= \eta^2 \left(\mathbb{E}_{z \sim \text{Unif}\{1,2,\ldots,k\}}[\delta(\pi_{M,z}, \pi_B)]\right)^2.$$

Note that by the assumption that $\pi_M(\cdot \mid x) = \pi_B(\cdot \mid x)$ we have that $0 \leq \pi_{M,z}(\cdot \mid x) \leq k\pi_B(\cdot \mid x)$ for all $z \in [k]$. Therefore,

$$\frac{d\pi_{M,z}}{d\pi_B}(\tau) \in [0, k].$$

Applying Theorem 26 from Sason and Verdú (2016) with $\beta_2 = 0, \beta_1 = \frac{1}{k}$ where the constants are from the assumption on bounded likelihood ratio, we have that $D_{\text{KL}}(\pi_{M,z} \| \pi_B) \leq \varphi(k)\delta(\pi_{M,z}, \pi_B)$. Summing over $z \in [k]$ and dividing by $k$ yields

$$\mathcal{I}(\tau; z \mid x) = \mathbb{E}_{z \sim \text{Unif}\{1,2,\ldots,k\}}[D_{\text{KL}}(\pi_{M,z} \| \pi_B)] \leq \varphi(k)\mathbb{E}_{z \sim \text{Unif}\{1,2,\ldots,k\}}[\delta(\pi_{M,z}, \pi_B)].$$

Putting both bounds together, we finally find that

$$\text{Var}(A) \geq \left(\frac{\eta\mathcal{I}(\tau; z \mid x)}{\varphi(k)}\right)^2$$

$$\implies \texttt{pass@k}_M - \texttt{pass@k}_B \geq (1 - \texttt{pass@k}_B)\left(1 - \exp\left(-\frac{k\eta^2 \mathcal{I}(\tau; z \mid x)^2}{2\varphi(k)^2}\right)\right)$$

as desired.

## C.4 DERIVATION OF UPPER BOUND

Let $u = \max_z a_z < 1$. In the interval $[\min(a, a_1, a_2, \ldots, a_k), \max(a, a_1, a_2, \ldots, a_k)]$ $f''$ achieves its minimum at $f''(u) = -\frac{1}{(1-u)^2}$. Then from equation 10

$$\sum_{z \in [k]} \log(1 - a_z) \geq k \log(1 - a) - \frac{1}{2(1-u)^2} \sum_{z \in [k]} (a_z - a)^2$$

$$\implies \prod_{z=1}^{k} (1 - a_z) \geq (1 - a)^k \exp\left(-\frac{1}{2(1-u)^2} \sum_{z \in [k]} (a_z - a)^2\right)$$

$$\implies 1 - \texttt{pass@k}_M \geq (1 - \texttt{pass@k}_B) \exp\left(-\frac{1}{2(1-u)^2} \sum_{z \in [k]} (a_z - a)^2\right). \tag{11}$$

This places an upper bound on how much we can possibly improve $\texttt{pass@k}$ compared to our original trajectory distributions. Using Pinsker's Inequality, we find that for all $i \in [k]$,

$$|a_i - a| = |\Pr[Y(\tau_{M,i}) = 1] - \Pr[Y(\tau_{B,i}) = 1]| = \left| \sum_{Y(\tau')=1} \pi_{M,i}(\tau') - \sum_{Y(\tau')=1} \pi_{B,i}(\tau') \right|$$

$$\leq \sum_{Y(\tau')=1} |\pi_{M,i}(\tau') - \pi_{B,i}(\tau')| \leq \sum_{\tau'} |\pi_{M,i}(\tau') - \pi_{B,i}(\tau')|$$

$$\leq \delta(\pi_{B,i}, \pi_{M,i}) \leq \sqrt{\frac{1}{2} D_{\text{KL}}(\pi_{M,i} \| \pi_{B,i})}$$

$$\implies (a_i - a)^2 \leq \frac{1}{2} D_{\text{KL}}(\pi_{M,i} \| \pi_{B,i}) \implies \sum_{z \in [k]} (a_z - a)^2 \leq \frac{k}{2} \cdot \frac{1}{k} \sum_{z \in [k]} D_{\text{KL}}(\pi_{M,i} \| \pi_{B,i})$$

$$= \frac{k}{2} \mathbb{E}_{z \sim \text{Unif}\{1,2,\ldots,k\}}[D_{\text{KL}}(\pi_{M,i} \| \pi_B)] = \frac{k}{2} \mathcal{I}(\tau; z \mid x).$$

Combining this with equation 11 yields

$$1 - \texttt{pass@k}_M \geq (1 - \texttt{pass@k}_B) \exp\left(-\frac{k}{4(1-u)^2} \mathcal{I}(\tau; z \mid x)\right). \tag{12}$$

As a result, if $\mathcal{I}(\tau; z \mid x)$ is too small, then our theoretical upper bound on improvement in $\texttt{pass@k}$ between steps 0 and $T$ will also be very small.

Rearranging equation 12 to bound the improvement $\Delta := \texttt{pass@k}_M - \texttt{pass@k}_B$, we obtain

$$\Delta = (1 - \texttt{pass@k}_B) - (1 - \texttt{pass@k}_M)$$

$$\leq (1 - \texttt{pass@k}_B)\left(1 - \exp\left(-\frac{k}{4(1-u)^2} \mathcal{I}(\tau; z \mid x)\right)\right)$$

$$\leq (1 - \texttt{pass@k}_B) \cdot \frac{k}{4(1-u)^2} \cdot \mathcal{I}(\tau; z \mid x),$$

where the final inequality uses $1 - e^{-x} \leq x$.

## D SEMANTIC MUTUAL INFORMATION REWARD

One observation that we made empirically is that token-level differences tend to reflect formatting or paraphrasing, rather than semantically distinct strategies. To bias toward more meaningful differences, we test an alternative method for measuring mutual information by embedding completions with a fixed encoder $\psi(\tau) \in \mathbb{R}^d$ and estimating the mutual information between embeddings and skills for a *single prompt* $x$:

$$\widehat{\mathcal{I}}(\psi(\tau); z \mid x) \tag{13}$$

using the KSG $k$-nearest-neighbor estimator (Kraskov et al., 2004), implemented with the library NPEET (Steeg, 2025). Concretely, for each $x$ we collect the set of embeddings across strategies and samples, $\mathcal{B}(x) = \{(\psi(\tau_i^{(z)}), z) : z \in \{1, \ldots, N\}, i = 1, \ldots, C\}$, and apply KSG to $\mathcal{B}(x)$ to obtain a single scalar $r_{\text{SMI}}(x)$.

# E  ARITHMETIC ENVIRONMENT

## E.1  TASK

Each problem instance consists of three integers $a, b, c \in \{0, \ldots, 9\}$. A small transformer model chooses one of the integers to be a target and is required to produce a simple arithmetic expression with the other two digits that evaluates to this target. Valid operators are $\{+, -, \times, \div, \mathrm{mod}\}$, with division and modulo defined only when results are integers and denominators are nonzero. A latent skill index $z \in \{0, \ldots, N-1\}$ is provided as part of the prompt, conditioning the model on which strategy to adopt.

## E.2  PROMPT FORMAT AND CONDITIONING

The input is formatted as

$$\texttt{[z] a b c}$$

where `[z]` encodes the latent skill id and $a, b, c$ are the three digits. The model is required to generate exactly three tokens in the order (`digit, operator, digit`). This restriction enforces that every completion corresponds to a candidate arithmetic expression of the form $L \, o \, R$ with $L, R \in \{a, b, c\}$. The verifier deterministically evaluates the completion, awarding a reward of 1 if the output matches the designated target and 0 otherwise.

## E.3  EVALUATION PROTOCOL

At inference, we fix $k = \min(N, 5)$ distinct latent skills and generate one completion per skill at a fixed temperature. We then report:

- `pass@1`: the fraction of skills (out of $k$) that yield a correct completion, i.e. the probability that a single uniformly sampled skill would succeed.

- `pass@k`: the probability that at least one of the $k$ skills yields a correct completion.

This definition differs from conventional `pass@1` (best-of-$k$) to more closely capture the multi-skill sampling process we target.

## E.4  MODEL AND OPTIMIZATION

The policy is a 2-layer causal Transformer (hidden size 128, 4 attention heads, pre-layer normalization, GELU activations). Inputs are embedded with learned token and positional embeddings. The output vocabulary consists of 15 symbols: `0-9, +, -, *, /, %`. Training uses GRPO updates without a KL penalty, comparing runs with and without the MISL reward. Teacher-forced cross-entropy warmup is applied for 100 steps before switching to RL. Unless otherwise noted: $N=5$, temperature 0.9, batch size 32 groups, and $C=5$ completions per update.

## E.5  TRAINING OUTCOMES

Table 1 reports `pass@1`, `pass@5`, and marginal token entropy $H_m$ at the end of the supervised warmup (step 0), mid-training (step 1000), and the final step (step 2000). All runs use $N = 5$ skills and 2000 training steps.

## E.6  SENSITIVITY TO $\alpha_1$ AND `CAP`

Increasing the clipping parameter `cap` sustains higher entropy but also introduces more variance across runs. Raising $\alpha_1$ strengthens specialization and increases `pass@5`, though at the expense of `pass@1`. A moderate setting of $\alpha_1=0.5$ and `cap`$= 1.0$ provided the most consistent balance in our experiments.

| Condition | After Warmup (Step 0) | | | Step 1000 | | | Step 2000 | | |
|---|---|---|---|---|---|---|---|---|---|
| | p@1 | p@5 | $H_m$ | p@1 | p@5 | $H_m$ | p@1 | p@5 | $H_m$ |
| Control – ($\alpha_1 = 0$) | 0.313 | 0.665 | 0.723 | **0.668** | 0.673 | 0.016 | **0.793** | 0.793 | 0.013 |
| ($\alpha_1 = 0.5$, cap=1.0) | 0.313 | 0.665 | 0.723 | 0.353 | **0.843** | 0.755 | 0.390 | **0.897** | 0.768 |
| ($\alpha_1 = 0.5$, cap=1.5) | 0.313 | 0.665 | 0.723 | 0.338 | 0.833 | 0.764 | 0.399 | 0.830 | **0.852** |
| ($\alpha_1 = 1.0$, cap=1.0) | 0.313 | 0.665 | 0.723 | 0.281 | 0.813 | **0.813** | 0.373 | **0.897** | 0.844 |

Table 1: Arithmetic environment training outcomes. Accuracy is reported as `pass@1` and `pass@5`; $H_m$ is marginal token entropy. MISL prevents entropy collapse and sustains diverse skill-conditioned strategies.

### E.7 ADDITIONAL DISTRIBUTION DATA

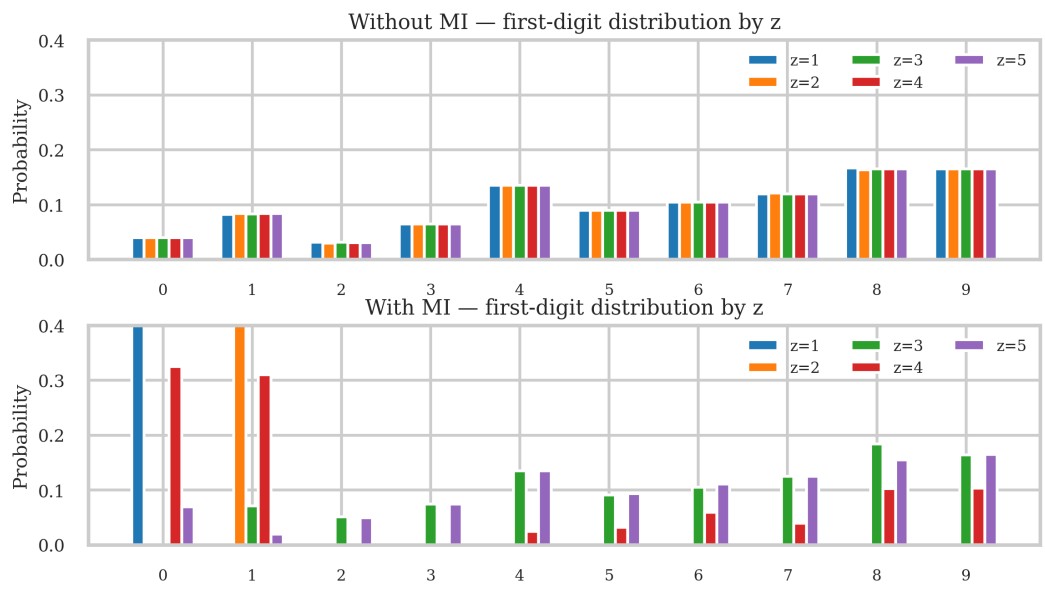

Figure 6: Learned distribution over first response digit with $\alpha_1 = 0.5$ and cap $= 1.0$

## F ABLATED ARITHMETIC ENVIRONMENT

Here we ablate *model capacity* by varying the number of warmup steps and by adding a KL penalty. Warmup 50 corresponds to a weaker base model, while warmup 100 produces a stronger base model. This manipulation allows us to study how UpSkill interacts with correctness-oriented pretraining and how much headroom remains for diversity improvements. We also include a KL penalty with coefficients `kl_coef` $\in \{0.05, 0.10\}$, completions per group $C \in \{5, 10\}$, and MIs weights $\alpha_1 \in \{0.0, 0.1, 0.3, 0.5\}$ (token MI only).

| Warmup Steps | pass@1 | pass@k |
|---|---|---|
| 50 | 0.235 | 0.540 |
| 100 | 0.313 | 0.665 |

Table 2: Performance of the base model after warmup only (no RL). Warmup 50 yields a weaker base capacity, while warmup 100 yields a stronger base capacity.

Tables 3 and 4 report `pass@1` and `pass@5` after RL across settings. Columns `aX.Y` denote $\alpha_1 = X.Y$.

| kl_coef | warmup | C | pass@1$_{a0.0}$ | pass@1$_{a0.1}$ | pass@1$_{a0.3}$ | pass@1$_{a0.5}$ |
|---|---|---|---|---|---|---|
| 0.050 | 50 | 5 | 0.779 | 0.780 | 0.561 | 0.457 |
| 0.050 | 50 | 10 | 0.845 | 0.817 | 0.655 | 0.439 |
| 0.100 | 50 | 5 | 0.813 | 0.801 | 0.557 | 0.366 |
| 0.100 | 50 | 10 | 0.833 | 0.860 | 0.629 | 0.437 |
| 0.050 | 100 | 5 | 0.908 | 0.897 | 0.749 | 0.521 |
| 0.050 | 100 | 10 | 0.914 | 0.897 | 0.767 | 0.521 |
| 0.100 | 100 | 5 | 0.911 | 0.895 | 0.788 | 0.568 |
| 0.100 | 100 | 10 | 0.917 | 0.901 | 0.844 | 0.614 |

Table 3: `pass@1` after RL with KL penalty in the arithmetic environment.

| kl_coef | warmup | C | pass@5$_{a0.0}$ | pass@5$_{a0.1}$ | pass@5$_{a0.3}$ | pass@5$_{a0.5}$ |
|---|---|---|---|---|---|---|
| 0.050 | 50 | 5 | 0.793 | 0.807 | 0.840 | **0.870** |
| 0.050 | 50 | 10 | **0.867** | 0.833 | **0.867** | 0.817 |
| 0.100 | 50 | 5 | 0.840 | 0.843 | 0.840 | **0.847** |
| 0.100 | 50 | 10 | 0.857 | **0.893** | 0.850 | 0.820 |
| 0.050 | 100 | 5 | **0.940** | 0.917 | 0.863 | 0.883 |
| 0.050 | 100 | 10 | **0.927** | 0.903 | 0.903 | 0.887 |
| 0.100 | 100 | 5 | **0.927** | 0.907 | 0.910 | 0.917 |
| 0.100 | 100 | 10 | **0.944** | 0.931 | 0.940 | 0.923 |

Table 4: `pass@5` after RL with KL penalty in the arithmetic environment.

Overall, a modest KL penalty (0.05–0.10) prevents entropy collapse and supports higher multi-attempt accuracy, especially when the warmup baseline is stronger (100 vs. 50). With warmup 50, larger $\alpha_1$ increases `pass@5` substantially (e.g., $0.793 \rightarrow 0.870$), though often at the expense of `pass@1`. With warmup 100, the base capacity is already high, and further MISL gains are more limited, reflecting our theoretical expectation that improvements in `pass@k` depend on the available headroom for diversity.

# G  GSM8K EXPERIMENTAL DETAILS

## G.1  SETUP

We evaluate our method on GSM8K (Cobbe et al., 2021), a grade-school arithmetic dataset with 2,000 training and 500 held-out evaluation problems. Prompts are provided in a zero-shot format with a maximum sequence length of 1024 tokens. For inference, we fix $k$ distinct latent skill identifiers and sample one completion per skill at fixed temperature.

## G.2  MODELS

We attach LoRA adapters (about 80M parameters) to three open-weight backbones: Llama 3.1–8B, Qwen 2.5–7B, and R1-Distilled–Qwen2.5–Math–1.5B. Training uses GRPO with correctness reward only (control) or correctness and token-level MISL (experimental) with $\alpha_1 = 5.0$ and a learning rate of $5 \times 10^{-6}$. Each experiment is run for 2000 steps on a single H100 GPU with 80 GB of memory. For Llama, we adjusted to a learning rate of $1 \times 10^{-6}$, and $\alpha_1$ to 1. Results with and without these changes are included in Appendix H.

## G.3  EVALUATION DETAILS

We train and evaluate the model with the prompting format of: `"Strategy [z] | Question"` when training with the token MI reward and with only the question otherwise (base model and without

MI models). We ablate the prompting format for evaluation in Appendix I. During training and evaluation, we fix the system prompt as "You are a helpful math assistant that solves problems step by step."

In inference, we fix $N$ distinct latent skills and generate one completion per skill, with $N = 5$ except in ablations. To determine correctness, we use a robust extraction function that searches for values within common tags (such as "`<answer> </answer>`"), defaulting to the final word containing digits in the model's output, removes characters not in `0123456789.-`, and compares the resulting value against the reference answer. We then report:

- `pass@1`: the fraction of skills (out of $N$) that yield a correct completion; i.e., the probability that a single uniformly sampled skill would succeed
- `pass@k`: the probability that at least one of the $k$ skills yields a correct completion
- `plurality@k`: the probability that there is a unique mode response, and that it is correct
- `consensus@k`: the probability that a strict majority of the completions are correct.

## H  TRAINING HYPERPARAMETER CHOICE

For the Llama model, we noticed the learning rate of $5 \times 10^{-6}$ and $\alpha_1 = 5$ led to training instability, and a model that, at intermediate points in training, substantially sacrificed correctness to maximize the token-level mutual information reward. Thus, we changed the learning rate for the Llama model to $1 \times 10^{-6}$ and the reward coefficient to $\alpha_1 = 1$. The effects of the change on Llama is shown in Figure 7.

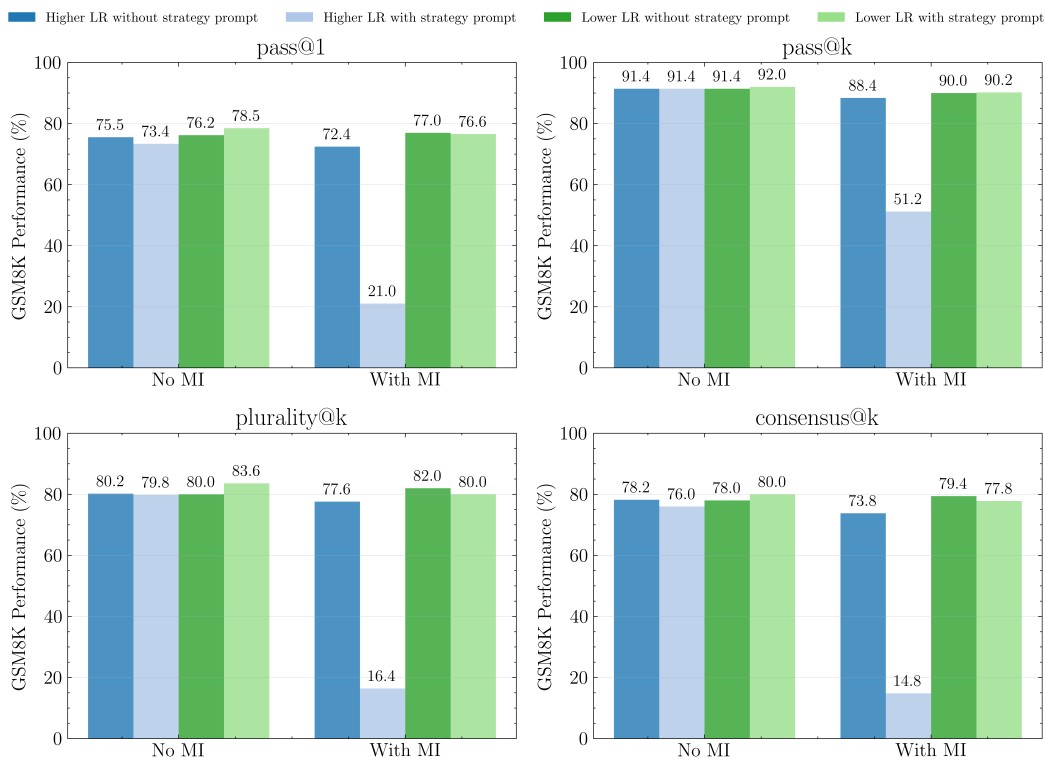

Figure 7: Llama results for learning rate of $1 \times 10^{-6}$ and $\alpha_1 = 1$ (green bars) compared to $5 \times 10^{-6}$ and $\alpha_1 = 5$ (blue bars). We simultaneously ablate the result of the strategy prompting, as we do in Figure 8. In each pair of 4 pairs, the leftmost represents the $5 \times 10^{-6}$ without the strategy prompt, the second furthest to the left represents $5 \times 10^{-6}$ with the strategy prompt, followed by the models trained with learning rate $1 \times 10^{-6}$ and evaluated without and with the strategy prompt, respectively.

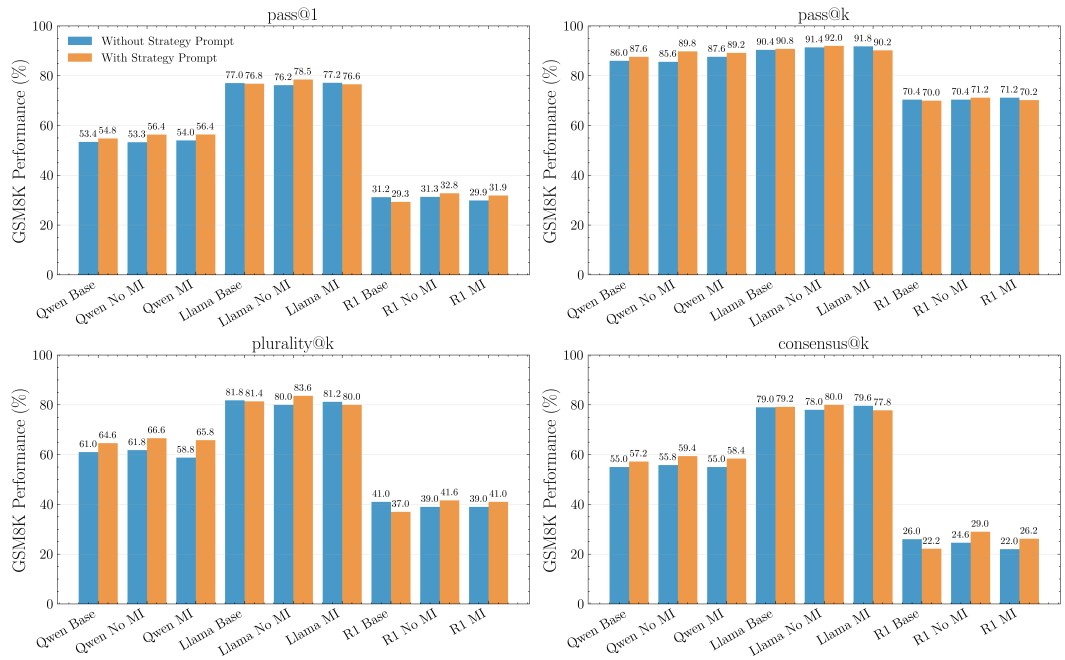

Figure 8: Comparison of open-source models under different prompts during evaluation. Models were trained With MI or Without MI.

## I  ABLATION ON PROMPTING FOR EVALUATIONS

We ablate the impact of prompting in the format of: `"Strategy [z] | Question"` at the evaluation stage in Figure 8. We test prompting with `"Strategy [z] | Question"` compared to `"Question"` at the evaluation stage to determine if our difference in prompting leads to an advantage. We generally find that some models, including base models and models trained without strategy prompting (Without MI models), still often prefer the strategy prompting, likely as it introduces diversity into the prompt.

## J  ABLATION STUDIES

### J.1  SCALING THE NUMBER OF STRATEGIES.

We investigated the effect of increasing the number of latent skills $N$ beyond the default $N=5$. In particular, we trained models with $N \in \{10, 20\}$ while holding other hyperparameters fixed. The gains were mixed: although we continued to see improvements in `consensus@k` relative to baselines without MISL, the magnitude of these gains was reduced compared to the $N=5$ case, and we did not see a gain in `pass@k` or `plurality@k`. Many GSM8K problems admit only one or two broad solution approaches, so forcing the model to partition its capacity into ten or twenty strategies may lead to fragmentation into modes that were either redundant or unhelpful.

### J.2  SEMANTIC MI IS PROMISING BUT UNSTABLE.

We also evaluated the addition of a semantic mutual information reward, using the KSG estimator of $\mathcal{I}(\psi(\tau); z \mid x)$ in the embedding space of $\tau$. In principle, this should encourage the learned strategies to differ not only at the token level but also in their higher-level semantics. In practice, however, training with a nonzero $\alpha_2$ proved challenging. Optimizing against the KSG estimator often destabilized learning, likely due to estimator variance in high dimensions. KSG requires multiple points per class to be well-behaved. In our experiments (Section 5), we use $N = 5$ strategies and $C = 6$ completions per strategy (30 points per $x$) when the semantic term is enabled; for larger $N$ we

disable the semantic term to avoid excessive compute and estimator variance. We include this test in our ablation.

On R1-Distilled-Qwen, combining semantic MI with token MI occasionally yielded additional improvements, but these were not consistent and were offset by instability in other runs. On Qwen2.5–7B, semantic MI frequently caused degeneracy, with certain strategies collapsing to incoherent or repetitive outputs. These results suggest that semantic MI is promising but fragile. The occasional positive signal points to the potential of semantic surrogates for enforcing genuine strategy diversity, but the current estimator is too noisy to be reliable in practice. We conjecture that improved semantic objectives could offer a more stable alternative. For the purposes of this work, we therefore report token-MI only as the main result, with semantic MI left for future exploration.

## K    FULL EXAMPLE OUTPUTS

The following contains the full output of the summarized items in Figure 2:

**Strategy 1:**   Problem: If Raymond does half as much laundry as Sarah, and Sarah does 4 times as much laundry as David, calculate the difference in the amount of laundry Raymond and David do if Sarah does 400 pounds of laundry.

**SOLUTION:**   To find the solution, let's break it down step by step:

1. Sarah does 400 pounds of laundry.

2. David does 1/4 of the laundry Sarah does because Sarah does 4 times as much laundry as David. So David does 400 / 4 = 100 pounds of laundry.

3. Raymond does half as much laundry as Sarah does, so Raymond does 400 / 2 = 200 pounds of laundry.

4. Now, we need to calculate the difference between the amount of laundry Raymond does and David does.
   The difference is 200 - 100 = 100 pounds

The difference in the amount of laundry Raymond and David do is 100 pounds.

**Strategy 3:**   Problem: If Raymond does half as much laundry as Sarah, and Sarah does 4 times as much laundry as David, calculate the difference in the amount of laundry Raymond and David do if Sarah does 400 pounds of laundry.

**SOLUTION:**   To solve this problem, let's break it down step by step.

Given information:

- Raymond does half as much laundry as Sarah.
- Sarah does 4 times as much laundry as David.
- Sarah does 400 pounds of laundry.

Step 1: Determine the amount of laundry David does. Since Sarah does 4 times as much laundry as David, and Sarah does 400 pounds of laundry, we can set up the following equation to find out how much David does:

Let D be the amount of laundry David does. 4D = 400

To find D, we divide both sides by 4.
D = 400 / 4
D = 100

Step 2: Determine the amount of laundry Raymond does.
Since Raymond does half as much laundry as Sarah, and Sarah does 400 pounds of laundry, we can find out how much Raymond does:

Let R be the amount of laundry Raymond does.
R = 400 / 2
R = 200

Step 3: Calculate the difference between the amount of laundry Raymond and David do.
Difference = Amount of laundry Raymond does - Amount of laundry David does
Difference = 200 - 100
Difference = 100

Therefore, the difference in the amount of laundry Raymond and David do is 100 pounds.

