# OpenReview forum: "UpSkill: Mutual Information Skill learning for Structured Response Diversity in LLMs"
_ICLR.cc/2026/Conference — Submitted to ICLR 2026_

### Official Review · Reviewer_ZGnJ · 2025-10-29

**Soundness:** 3
**Presentation:** 2
**Contribution:** 3
**Rating:** 4
**Confidence:** 3

**Summary:**

The paper addresses the critical issue of "diversity collapse" in LLMs, where standard reinforcement learning (RLVR) techniques aimed at optimizing single-attempt accuracy (pass@1) inadvertently reduce response variety, thereby limiting multi-attempt (pass@k) performance. The authors introduce UpSkill, a novel training-time method that integrates Mutual Information Skill Learning (MISL) into the GRPO framework. This is achieved by conditioning the model on a discrete latent "strategy" variable $z$ and introducing a token-level mutual information reward ($r_{TMI}$). This reward incentivizes the model to produce distinct, reproducible strategies for each $z$. Experiments demonstrate the method's effectiveness, starting with a controlled arithmetic environment where UpSkill prevents diversity collapse and significantly boosts pass@5, and extending to the GSM8K benchmark, where it achieves median gains in pass@k and consensus@k across several models without degrading pass@1 accuracy.

**Strengths:**

Basically, in my opinion, this paper's primary strength lies in its principled approach, tightly coupling the proposed method with a strong theoretical justification.
The authors provide a formal proof (Lemma 1) that directly links the mutual information objective $\mathcal{I}(\tau;z|x)$ to a bound on pass@k improvements, moving the work beyond simple empirical heuristics.
Moreover, the theoretical proof is well-supported by the empirical results. The "arithmetic environment" (Section 5.1, Figure 4) serves as an good controlled experiment, clearly visualizing how the baseline model's diversity completely collapses (pass@1 converges to pass@5) while UpSkill maintains a significant performance gap, proving the concept. The method addresses a good implement potential, showing that on the complex GSM8K task, it is possible to enhance multi-attempt metrics (pass@k, consensus@k) without the common side effect of sacrificing single-attempt (pass@1) performance.

**Weaknesses:**

Basically, I think the experiments and analysis should be more sufficient and solid.
1） The experiment results for the Llama 3.1-8B model is decently negative and seems negative, where pass@k performance decreased by 2% (from 88% to 86%). The lack of discussion on this contradictory finding in the main text undermines the method's claimed robustness, especially on high-capability models.

2）The presentation of the main GSM8K results is incomplete. The specific value of $k$ used for the pass@k, plurality@k, and consensus@k metrics is not explicitly stated in the main body of the paper, making the magnitude of the improvements difficult to interpret.

3) The main results chart (Figure 5) is missing a numerical label for the Llama 3.1-8B pass@1 "with MI" condition, which prevents a full and clear comparison of this key metric on the strongest baseline model.

**Questions:**

1) Could the authors elaborate on the negative pass@k result for Llama 3.1-8B? Does this finding suggest a fundamental limitation of UpSkill when applied to stronger base models, perhaps aligning with the findings in Appendix F, or is it potentially an artifact of hyperparameter selection?

2) What is the computational overhead of the token-level MI reward (Equation 5) during training? Does calculating the mixture distribution $p_{\pi}(y_t|x,y_{<t})$ require N separate forward passes, and if so, how does this impact training efficiency?

3) Given that other contemporary work (e.g., Chen et al., 2025) explores the direct optimization of the pass@k objective, what is the primary advantage of using the mutual information approach? Does UpSkill, for instance, produce more semantically distinct or interpretable strategies than direct pass@k optimization would?

---

> ### Author Response · Authors · 2025-11-28
>
> Dear Reviewer,
>
> Thank you for the detailed feedback for improving the paper, and we address all points below. We apologize for the delayed response time, it required substantial time to conduct the additional experimentation.
>
> **[Additional Experimentation]**
>
> We conduct a new series of experiments on an OOD coding dataset HumanEval [1] and ablate for the impact of GRPO pass@1 correctness. For length, we include abridged results here and full results in the response to Reviewer CCs5.
>
> To avoid confounding the impact of GRPO pass@1 training and UpSkill, we perform the following experiments without pass@1 correctness training and test an optional KL penalty ($D_{\text{KL}}(\pi_M(\cdot\mid x)\parallel\pi_B(\cdot\mid x))$). This isolates the contribution of the token-level mutual information reward, highlights the distinction between UpSkill and correctness-labeled pass@k approaches, and examines training stability under the KL constraint.
>
> Below we provide the results from Qwen and Llama models, showing the impact of UpSkill with solely a token-level MI reward (MI) and with the token-level MI reward and KL penalty (MI + KL).  As noted, these models are trained without any correctness reward. The results of the R1 model are not included here, but it is the smallest model, at 1.5B parameters, and decreased in performance with UpSkill.
>
> ### **Qwen (GSM8K)**
>
> | Method               | pass@1 | pass@k | plurality@k | consensus@k | Δ pass@k |
> | -------------------- | ------ | ------ | ----------- | ----------- | -------- |
> | **BASE**             | 53.4   | 86.0   | 61.0        | 55.0        | **0.0**  |
> | **MI**                     | 61.6   | 91.2   | 72.0        | 65.6        | **+5.2** |
> | **MI + KL (low lr)** | 55.0   | 90.0   | 67.4        | 45.6        | **+4.0** |
>
>
> We again observe that the MI reward alone increases pass@k, and surprisingly also increases pass@1 despite the absence of a correctness reward.
>
>
> ### **Llama (GSM8K)**
>
> | Method               | pass@1 | pass@k | plurality@k | consensus@k | Δ pass@k |
> | -------------------- | ------ | ------ | ----------- | ----------- | -------- |
> | **BASE**             | 77.0   | 90.4   | 81.8        | 79.0        | **0.0**  |
> | **MI**               | 75.2   | 91.2   | 80.8        | 77.4        | **+0.8** |
> | **MI + KL (low lr)** | 77.64  | 92.8   | 81.4        | 79.0        | **+2.4** |
>
> ---
>
> For Llama, both MI and MI + KL improve pass@k, with the KL penalty providing the strongest improvement.
>
> ---
>
> We additionally evaluate out-of-distribution performance by applying GSM8K-trained UpSkill models to HumanEval.
>
> ### **Qwen (HumanEval)**
>
> | Method                | pass@1 | pass@5 | Δ pass@5 |
> | --------------------- | ------ | ------ | -------- |
> | **BASE**              | 68.8   | 80.0   | **0.0**  |
> | **MI + KL (low lr)**           | 70.7   | 82.9   | **+2.9** |
> | **MI + KL (high lr)** | 60.3   | 84.1   | **+4.1** |
>
>
> For Qwen, UpSkill improves OOD pass@5 performance.
>
> ### **Llama (HumanEval)**
>
> | Method                | pass@1 | pass@5 | Δ pass@5 |
> | --------------------- | ------ | ------ | -------- |
> | **BASE**              | 57.3   | 72.5   | **0.0**  |
> | **MI + KL (low lr)**           | 54.1   | 67.7   | **–4.8** |
> | **MI + KL (high lr)** | 45.0   | 65.0   | **–7.5** |
>
> However UpSkill did not improve Llama performance. It is possible that behaviors Llama learned with respect to GSM8k do not translate as well to HumanEval.
>
> ---
>
> Full tables and experimental details are included in our response to Reviewer CCs5. These results show improvements for Llama and Qwen in pass@k on GSM8k and improvements for the Qwen model on the OOD dataset. Additionally, we note that different z correspond to different behavior as predicted, with some especially distinctive behaviors including responding in a different language, responding with extreme verbosity, and going through the problem narratively.
>
> **[W1]** We find that improved performance is due to increased response diversity, with different z conditioning the model to respond in different ways. Some modes lead to a drop in performance, as illustrated in the example for reviewer CCs5, and others perform strongly, overall leading to an improvement in pass@k at a cost of pass@1. However, challenges in UpSkill training derive from balancing the UpSkill objective with ensuring the model does not engage in catastrophic forgetting. As we standardized the Figure 5 training hyperparameters across models, the Figure 5 results involved a cautious training setting for Llama, leading to only minor performance changes and a more minor contribution from the UpSkill objective.

---

> > ### Author Response · Authors · 2025-11-28
> >
> > **[W2/W3]** We now clearly state that k = 5 and have improved the figure captions.
> >
> > **[Q2]** During training, the token-level MI reward currently requires k forward passes per training step, as each individual skill must be queried once to compute the log-probs of the trajectory for that distribution. Rather than one forward and one backward pass, instead we must perform O(k) forward passes and one backward pass. For large k we find that each UpSkill training step is approximation k/3 times longer than a standard GRPO training step; with k=0 (i.e. standard GRPO) each iteration takes about 30 seconds per step on average, with k=5 this is 40 seconds and with k=20 this is 250 seconds.
> >
> > **[Q3]** We see UpSkill as orthogonal work to Chen et al. and other contemporary work. While Chen et al. directly optimizes for the pass@k objective with a sophisticated sampling method over rollouts, UpSkill’s mutual information loss explicitly introduces multiple strategies for the model that are semantically distinct from each other in a replicable manner. Moreover, as demonstrated above, UpSkill does not require correctness labels to be effective, a distinguishing characteristic from Chen et al. Chen et al. mention that their work is supplementary to entropy-guided approaches, such as UpSkill, in Sec. 3.1 of their paper [2]. The two methods serve different purposes and thus can be combined to enhance pass@k performance.
> >
> > **[Figure 5 Paper Changes]**
> >
> > Upon review, we realized that our model improvement in earlier experiments was largely due to improving adherence to parsing requirements due to an overtly strict extraction function, and that we had erred in the Qwen GSM8k evaluation. We have adjusted the paper to include the new more robust parser, which we describe in Appendix G, and fix the other evaluation issues. The new experiments are run with the more robust parser. We are very sorry for these issues and are now confident the new results represent improvements in the underlying task.
> >
> > **[Conclusion]**
> > We apologize again for the delay in experimentation. We hope that these experiments, together with the additional revisions and clarifications discussed above, fully address the reviewers feedback about the paper, and we are happy to provide additional clarification if needed.
> >
> > Thank you,
> >
> > Authors
> >
> > --
> >
> > [1] Mark Chen et al. Evaluating Large Language Models Trained on Code, 2021.
> > URL https://arxiv.org/abs/2107.03374.
> >
> > [2] Zhipeng Chen, Xiaobo Qin, Youbin Wu, Yue Ling, Qinghao Ye, Wayne Xin Zhao, and Guang
> > Shi. Pass@k training for adaptively balancing exploration and exploitation of large reasoning
> > models, 2025. URL https://arxiv.org/abs/2508.10751.

---

### Official Review · Reviewer_CCs5 · 2025-10-30

**Soundness:** 1
**Presentation:** 2
**Contribution:** 2
**Rating:** 2
**Confidence:** 4

**Summary:**

The paper introduces UpSkill, a method to enhance "structured response diversity" in LLMs through Mutual Information Skill Learning (MISL). By conditioning on a discrete latent variable *z*, the model learns distinct reasoning strategies, which impoves multi-attempt metrics (on the arithmetic environment and gsm8k) while preserving single-attempt accuracy (only on gsm8k).

**Strengths:**

This paper presents a comprehensive theoretical analysis aimed at establishing a connection between mutual information and the llm's pass@K bounds. By introducing token-level mutual information into the reward, the approach enhances the diversity of the model's output strategies, thereby increasing the success rate of Pass@K. The proposed idea is novel enough, and experimental results indicate that it effectively improves the model's Pass@K capability.

**Weaknesses:**

1. The experiment results of the paper are not convincing. First, the results across the two different experimental settings are inconsistent: in the first experiment, Pass@K improves but Pass@1 stays low, while in the second experiment, Pass@K improves without degrading Pass@1. This indicates that the results are highly sensitive to the environment/dataset, but the paper only tests this method on two environments/datasets, which makes the results lack robustness. Moreover, the experiments only include in-distribution evaluations, without any out-of-distribution (OOD) benchmarks, making it difficult to see the generalization of this approach. Additionally, the analysis of the experiment section is also not enough to explain the results.
2. In the proof section, the paper introduces a strong assumption, but later states that this assumption is difficult to hold in practice. As a result, despite extensive theoretical analysis, the theory does not sufficiently support the final experimental results.
3. Finally, the contribution of the approach remains unclear. Although this approach may be beneficial for improving Pass@K, its performance on Pass@1 is unstable, and it requires careful tuning of the number of *z* (which significantly affects the performance and stability). Moreover, while *z* can control the diversity of responses, the actual meanings of different *z*s are not interpretable, making it nearly impossible for users to use *z* to control the strategies in llm's solutions.

**Questions:**

1. Have you analyzed whether the learned *z* acquires any generalizable semantic meaning after training?
2. Have you attempted to evaluate the method's generalization ability across different domains?
3. The experimental analysis in gsm8k explains that the pass@1 is maintained due to the existence of a larger set of correct
reasoning approaches. But later in the ablation part, it is mentioned that many GSM8K problems admit only a limited number
of distinct solution paths. There seems to be a contradiction between these statements, and this further strengthens my concern about the generalizability of *z*.
4. The analysis "UpSkill improvement is negatively related to pass@1 (equivalently
pass@kB ) capability" which explains the low Pass@1 in the Arithmetic Environment seems to conflict with the observation in gsm8k, where Pass@1 is maintained or even improved. Could you please provide an analysis to clarify this discrepancy?
5. During testing, if only a single response is generated, how should one select *z* to ensure the quality of the response? Is it possible to let the model choose *z* automatically, rather than requiring manual specification?

---

> ### Author Response · Authors · 2025-11-28
>
> Dear Reviewer,
>
> Thank you for your thorough and constructive feedback and we appreciate the abundance of detail. We are sorry for the delay in posting our rebuttal, it required substantial time to conduct the additional experimentation. Below we will address each point:
>
> **[Q1]** We have found that UpSkill leads to distinguishable performance. For the arithmetic environment (Section 5.1), we have found that z correlates with both the operators used (Figure 4) and with the numbers attempted (Figure 6). As an example, z = 5 solely corresponds to attempting division and modular arithmetic with large numbers. In the language environment, z often corresponds to distinguishable modes. For a Llama model trained with MISL reward (lr = 1e-4, alpha_1 = 5.0),  we observed that:
> - Generations with z = 1 were often self-referential and chaotic in responses (54.6% pass@1)
> - Generations with  z = 2 were often clean and offered step-by-step solutions (75.2% pass@1).
> - Generations with z = 3, were often more narrative, beginning with a repetition of the problem statement (79.2% pass@1).
> - Generations with z = 4 were often verbose and very lengthy in their explanations (76.4% pass@1).
> - Generations with z = 5 were similar to those with z = 2 (72.2% pass@1).
>
> For a separate Qwen model we analyzed, modes were different, and some notable behaviors we noticed included responding in Spanish, responding in Chinese, and responding in programming syntax.
>
> **[W1]** This approach does not require careful tuning of z for strong performance. Notably, for pass@k, we suggest to use |Z| = k, and our experiments are performed with |Z| = 5, without any tuning of |Z|. For model training, our only recommendation is to leverage a sufficient learning rate so that token-level MI increases,
>
> **[W2]** While enforcing the constraint in a strict fashion is impractical, it is possible to approximate the constraint $\pi_M(\cdot\mid x)=\pi_B(\cdot\mid x)$ by adding a reward estimating the negative KL divergence $D_{\text{KL}}(\pi_M(\cdot\mid x)\parallel\pi_B(\cdot\mid x))$ between the mixture trajectory distribution and the base model trajectory distribution. To showcase this technique leads to strong models, we have trained models using both our mutual information reward and this negative KL reward in the Additional Experimentation section. We find that this penalty, which encourages adherence to the theoretical requirements, still leads to strong performance improvements for pass@k.
>
> **[W3/Q5]** Although we agree that the interpretation of the different z-conditioned policies is challenging, the intent of UpSkill is not necessarily to create interpretable strategies, which can be done more efficiently by prompting the model directly to use specific strategies, such as asking the model to use algebraic, geometric, or combinatorial methods on a difficult math problem. Rather, we intend for these strategies to naturally differentiate from each other in a generalizable manner, so that the model can be easily queried for k different strategies on any problem without needing any domain-specific knowledge of what strategies would work the best.
>
> During inference, if only one response from the model is desired, one can sample from the unconditioned trajectory distribution by first sampling a skill uniformly at random from the possible skills (with k=5 picking a uniformly random number 1 through 5), then sampling one trajectory from the model conditioned on that skill. While the model does not choose the skill itself, the only additional overhead required is the random sampling of the skill variable.
>
> **[Q3]** In the ablation, we consider |Z| = 10 and |Z| = 20, whereas in the main paper we consider |Z| = 5, and thus both statements can be correct.  Increasing |Z| corresponds to increased diversity at the cost of training instability and so does not consistently lead to improved pass@k, especially when new modes are ineffective.
>
> **[Q4]** Thank you for the feedback here - we agree that this response was confusing. Our conjectured reason why the pass@1 is much lower in the arithmetic environment as compared to GSM8K is that the model’s final answer is much more sensitive to changes in trajectories in the former environment. If any of the three output tokens in the arithmetic setting are modified, the result of the evaluated expression will almost always be different. In contrast, in the textual reasoning setting, the model can significantly change its reasoning methods without necessarily arriving at a different final answer. Therefore, the model is able to maximize the mutual information reward in the second setting without necessarily negatively affecting the pass@1. Additionally, the introduction of variance from UpSkill can improve GRPO pass@1 training, providing a challenging confounding variable. We introduce experiments to isolate the impact of UpSkill in the Additional Experiments section. The experiments show a pass@1 improvement in Qwen.

---

> ### Author Response · Authors · 2025-11-28
>
> **[Additional Experiments]**
>
> We perform a series of additional experiments to test the impact of UpSkill without the confounding variable of simultaneous pass@1 training. We train models with the token-level mutual information reward and without any correctness reward, thus not requiring answer labels for data. This provides a distinguishing characteristic between UpSkill and other pass@k methods, as UpSkill does not require answer-labeled domain data. We additionally examine the impact of a KL constraint that ensures the UpSkill policy adheres closer to the requirements of Theorem 1, and we examine out of distribution performance. In summary, we test:
>
> - UpSkill impact independent of pass@1 correctness reward, a distinguishing feature of UpSkill.
> - OOD performance by testing models trained on GSM8k on the HumanEval [1] coding dataset.
> - The performance under a KL constraint motivated by the theory.
>
> We find that, for the Qwen and Llama models, UpSkill improves pass@k performance on in-distribution tasks, and for the Qwen model, UpSkill additionally improves pass@k performance on out of distribution tasks. Additionally, we find that Upskill improves performance even without labeled answers, a distinguishing characteristic from other pass@k improvement approaches. In the below experiments, Base refers to the performance of a base model, MI refers to the performance of a model with token-level mutual information rewards, and MI + KL refers to a model with token-level MI rewards and a reward encouraging adherence to the KL constraint (see [W2]). The R1 model is the smallest model, at 1.5B parameters, and tended to decrease in performance with UpSkill. We hypothesize this is due to weaker base model performance corresponding to greater training instability.
>
> We give specific hyperparameters below the experiments.
>
>
> ### **GSM8K**
>
> ---
>
> ### **Qwen (GSM8K)**
>
> | Method               | pass@1 | pass@k | plurality@k | consensus@k | Δ pass@k |
> | -------------------- | ------ | ------ | ----------- | ----------- | -------- |
> | **BASE**             | 53.4   | 86.0   | 61.0        | 55.0        | **0.0**  |
> | **MI**                     | 61.6   | 91.2   | 72.0        | 65.6        | **+5.2** |
> | **MI + KL (low lr)** | 55.0   | 90.0   | 67.4        | 45.6        | **+4.0** |
>
> We again observe that the MI reward alone increases pass@k, and surprisingly also increases pass@1 despite the absence of a correctness reward.
>
> ---
>
> ### **Llama (GSM8K)**
>
> | Method               | pass@1 | pass@k | plurality@k | consensus@k | Δ pass@k |
> | -------------------- | ------ | ------ | ----------- | ----------- | -------- |
> | **BASE**             | 77.0   | 90.4   | 81.8        | 79.0        | **0.0**  |
> | **MI**               | 75.2   | 91.2   | 80.8        | 77.4        | **+0.8** |
> | **MI + KL (low lr)** | 77.64  | 92.8   | 81.4        | 79.0        | **+2.4** |
>
> For Llama, both MI and MI + KL improve pass@k, with the KL penalty providing the strongest improvement.
>
> ---
>
> ### **R1 (GSM8K)**
>
> | Method               | pass@1 | pass@k | plurality@k | consensus@k | Δ pass@k |
> | -------------------- | ------ | ------ | ----------- | ----------- | -------- |
> | **BASE**             | 31.2   | 70.4   | 41.0        | 26.0        | **0.0**  |
> | **MI**               | 21.8   | 62.0   | 25.4        | 12.8        | **–8.4** |
> | **MI + KL (low lr)** | 30.24  | 71.2   | 37.4        | 25.0        | **+0.8** |
>
> For R1, MI substantially decreases pass@1 and pass@k.
>
> ---
>
> ### **HumanEval**
>
> ---
>
> ### **Qwen (HumanEval)**
>
> | Method                | pass@1 | pass@5 | Δ pass@5 |
> | --------------------- | ------ | ------ | -------- |
> | **BASE**              | 68.8   | 80.0   | **0.0**  |
> | **MI + KL (low lr)**           | 70.7   | 82.9   | **+2.9** |
> | **MI + KL (high lr)** | 60.3   | 84.1   | **+4.1** |
>
> For Qwen, UpSkill improves OOD pass@5 performance.
>
> ---
>
> ### **Llama (HumanEval)**
>
> | Method                | pass@1 | pass@5 | Δ pass@5 |
> | --------------------- | ------ | ------ | -------- |
> | **BASE**              | 57.3   | 72.5   | **0.0**  |
> | **MI + KL (low lr)**           | 54.1   | 67.7   | **–4.8** |
> | **MI + KL (high lr)** | 45.0   | 65.0   | **–7.5** |
>
> However UpSkill did not improve Llama performance. It is possible that behaviors Llama learned with respect to GSM8k do not translate as well to HumanEval.
>
> ---
>
> ### **R1 (HumanEval)**
>
> | Method                | pass@1 | pass@5 | Δ pass@5  |
> | --------------------- | ------ | ------ | --------- |
> | **BASE**              | 28.0   | 56.6   | **0.0**   |
> | **MI + KL (low lr)**           | 26.1   | 54.8   | **–1.8**  |
> | **MI + KL (high lr)** | 0  | 0        | **–** |
>
> Training failed for MI + KL (high lr)  due to the high learning rate.

---

> ### Author Response · Authors · 2025-11-28
>
> Hyperparameters
>
> The MI models were trained with $\alpha_1 = 5.0$, no correctness reward, and a learning rate of $1\times 10^{-4}$, no KL divergence penalty from the base model was applied. The Llama MI + KL models were trained with $\alpha_1 = 1.0$, $\alpha_{\text{KL}} = 0.2$, and no correctness reward, while the Qwen and R1 MI + KL models were trained with $\alpha_1 = 5.0$, $\alpha_{\text{KL}} = 0.5$, and no correctness reward; these hyperparameters were roughly set to make the two reward terms roughly equal in magnitude to each other. The learning rates for the low LR MI + KL models were $3\times 10^{-6}$ for Llama and $5\times 10^{-6}$ for Qwen and R1, while the learning rate for the high LR MI + KL models was $1\times 10^{-4}$.
>
> For the MI models, we leverage early stopping of training at the checkpoint with highest mutual information as, due to the high learning rate, training is occasionally unstable. We take a checkpoint every 100 steps out of the 2000 training steps.
>
> **[Figure 5 Paper Changes]**
>
> Upon review, we realized that our model improvement in earlier experiments was largely due to improving adherence to parsing requirements due to an overtly strict extraction function, and that we had erred in the Qwen GSM8k evaluation. We have adjusted the paper to include the new more robust parser, which we describe in Appendix G, and fix the other evaluation issues. The new experiments are run with the more robust parser. We are very sorry for these issues and are now confident the new results represent improvements in the underlying task.
>
> **[Conclusion]**
>
> We appreciate your concerns, and we apologize for the time needed for the above experiments. We hope these experiments and the above clarification address the reviewer’s concerns, and we are more than happy to provide additional clarification if needed.
>
> Thank you,
>
> Authors
>
>
> [1] Mark Chen et al. Evaluating Large Language Models Trained on Code, 2021.
> URL https://arxiv.org/abs/2107.03374.

---

### Official Review · Reviewer_Vey4 · 2025-11-03

**Soundness:** 3
**Presentation:** 3
**Contribution:** 3
**Rating:** 8
**Confidence:** 2

**Summary:**

This paper addresses the diversity collapse problem in RLVR, where existing methods like GRPO, which optimize for single-attempt accuracy (pass@1), can inadvertently reduce output diversity. The authors propose UpSkill, a training-time method that conditions the model on a latent discrete variable z and employs a novel token-level mutual information reward to encourage distinct responses. Experiments on GSM8K demonstrate that UpSkill can improve pass@k and consensus@k metrics without degrading pass@1 accuracy. The method is also supported by a theoretical analysis linking the mutual information objective to a lower bound on pass@k improvement.

**Strengths:**

1. The paper targets a crucial problem in existing RL training: the trade-off between single-attempt accuracy and multi-attempt diversity. The authors proposed a novel adaptation of MISL, and the formulation of the token-level mutual information reward made a valuable contribution.
2. The proposed method is supported by theoretical analysis, which links the mutual information objective to a lower bound on pass@k improvement.
3. Experiments show that UpSkill is effective at improving both the single-attempt accuracy and multi-attempt diversity.

**Weaknesses:**

1. The scalability of UpSkill appears limited, as its experiments are confined to the GSM8K dataset with N=5, a relatively simple benchmark for mathematical reasoning.
2. The method's stability is questionable, as it shows inconsistent effects on the pass@1 metric across different arithmetic and GSM8K tasks. It also remains unclear whether UpSkill is effective when applied to more powerful base models.

**Questions:**

1. What if we experiment with UpSkill on more complicated reasoning tasks? I would like to see the results of whether UpSkill can still **effectively** improve the multi-attempt diversity when the query is inherently complex (this would potentially yield more diverse solutions).

---

> ### Author Response · Authors · 2025-11-28
>
> Dear Reviewer,
>
> Thank you for your time and your feedback. We are sorry for the delay in posting our rebuttal, it required substantial time to conduct the additional experimentation to better show the benefits of UpSkill. We conduct a new series of experiments on an OOD coding dataset HumanEval [1] and ablate for the impact of GRPO pass@1 correctness. For length, we include abridged results here and full results in the response to Reviewer CCs5.
>
> To avoid confounding the impact of GRPO pass@1 training and UpSkill, we perform the following experiments without pass@1 correctness training and test an optional KL penalty ($D_{\text{KL}}(\pi_M(\cdot\mid x)\parallel\pi_B(\cdot\mid x))$). This isolates the contribution of the token-level mutual information reward, highlights the distinction between UpSkill and correctness-labeled pass@k approaches, and examines training stability under the KL constraint.
>
> Below we provide the results from Qwen and Llama models, showing the impact of UpSkill with solely a token-level MI reward (MI) and with the token-level MI reward and KL penalty (MI + KL).  As noted, these models are trained without any correctness reward. The results of the R1 model are not included here, but it is the smallest model, at 1.5B parameters, and decreased in performance with UpSkill.
>
> ### **Qwen (GSM8K)**
>
> | Method               | pass@1 | pass@k | plurality@k | consensus@k | Δ pass@k |
> | -------------------- | ------ | ------ | ----------- | ----------- | -------- |
> | **BASE**             | 53.4   | 86.0   | 61.0        | 55.0        | **0.0**  |
> | **MI**                     | 61.6   | 91.2   | 72.0        | 65.6        | **+5.2** |
> | **MI + KL (low lr)** | 55.0   | 90.0   | 67.4        | 45.6        | **+4.0** |
>
>
> We again observe that the MI reward alone increases pass@k, and surprisingly also increases pass@1 despite the absence of a correctness reward.
>
>
> ### **Llama (GSM8K)**
>
> | Method               | pass@1 | pass@k | plurality@k | consensus@k | Δ pass@k |
> | -------------------- | ------ | ------ | ----------- | ----------- | -------- |
> | **BASE**             | 77.0   | 90.4   | 81.8        | 79.0        | **0.0**  |
> | **MI**               | 75.2   | 91.2   | 80.8        | 77.4        | **+0.8** |
> | **MI + KL (low lr)** | 77.64  | 92.8   | 81.4        | 79.0        | **+2.4** |
>
> ---
>
> For Llama, both MI and MI + KL improve pass@k, with the KL penalty providing the strongest improvement.
>
> ---
>
> We additionally evaluate out-of-distribution performance by applying GSM8K-trained UpSkill models to HumanEval.
>
> ### **Qwen (HumanEval)**
>
> | Method                | pass@1 | pass@5 | Δ pass@5 |
> | --------------------- | ------ | ------ | -------- |
> | **BASE**              | 68.8   | 80.0   | **0.0**  |
> | **MI + KL (low lr)**           | 70.7   | 82.9   | **+2.9** |
> | **MI + KL (high lr)** | 60.3   | 84.1   | **+4.1** |
>
>
> For Qwen, UpSkill improves OOD pass@5 performance.
>
> ### **Llama (HumanEval)**
>
> | Method                | pass@1 | pass@5 | Δ pass@5 |
> | --------------------- | ------ | ------ | -------- |
> | **BASE**              | 57.3   | 72.5   | **0.0**  |
> | **MI + KL (low lr)**           | 54.1   | 67.7   | **–4.8** |
> | **MI + KL (high lr)** | 45.0   | 65.0   | **–7.5** |
>
> However UpSkill did not improve Llama performance. It is possible that behaviors Llama learned with respect to GSM8k do not translate as well to HumanEval.
>
> ---
>
> Full tables and experimental details are included in our response to Reviewer CCs5. These results show improvements for Llama and Qwen in pass@k on GSM8k and improvements for the Qwen model on the OOD dataset. Additionally, we note that different z correspond to different behavior as predicted, with some especially distinctive behaviors including responding in a different language, responding with extreme verbosity, and going through the problem narratively.
>
> Separately, upon review, we realized that our model improvement in the earlier paper GSM8k experiments was largely due to improving adherence to parsing requirements due to an overtly strict extraction function, and that we had erred in the Qwen GSM8k evaluation. We have adjusted the results paper to include the new more robust parser, which we describe in Appendix G, and fix the other evaluation issues. The new experiments are run with the more robust parser. We are very sorry for these issues and are now confident the new results represent improvements in the underlying task.
>
> We appreciate the positive feedback from the reviewer and apologize again for the delay in experimentation. We are more than happy to clarify any additional concerns if needed.
>
> Thank you,
>
> Authors
>
> [1] Mark Chen et al. Evaluating Large Language Models Trained on Code, 2021.
> URL https://arxiv.org/abs/2107.03374.

---

### Author Response · Authors · 2025-12-03
**Summary of Reviewer Feedback and Our Responses**

Dear AC,

We thank the AC and reviewers for the constructive feedback and the opportunity to clarify our work. We use this section to highlight some of the topics raised throughout the rebuttal period.

Reviewer feedback generally fell into three categories: (1) insufficient adherence to theoretical assumptions, (2) concerns about training instability and generalizability to out-of-distribution performance, and (3) questions about the interpretability and uses of the resulting skills.

In our paper, we ran experiments by introducing a latent skill variable and fine-tuning Qwen 2.5-7B, Llama 3.1-8B, and R1-Distilled-Qwen2.5-Math-1.5B with GRPO, using both the traditional correctness reward as well as a token-level mutual information (MI) reward that encourages response diversity (over all skills) while still associating each skill with specific reproducible behaviors.

To address the first concern on theory, we introduce a negative KL estimator reward (KL) that forces the model’s outputs to approximately satisfy the theoretical constraint. For our new experiments included below, we ablate the impact of the KL reward and find strong performance with and without the reward.

To address concerns on training instability and OOD generalization, we first corrected our GSM8K verifier and evaluation code. With these fixes, earlier reviewer concerns about pass@1 instability are greatly diminished. Next, we have performed new experiments to isolate UpSkill’s contribution from the impact of a correctness reward, and find that, when training on GSM8K with k=5:

- Qwen with MI improves pass@k by 5.2% and MI + KL improves pass@k by 4.0%.
- Llama with MI improves pass@k by 0.8% and MI + KL improves pass@k by 2.4%

In OOD evaluations, testing models trained on GSM8k on the HumanEval coding benchmark,
- Qwen with MI + KL improves between 2.9% and 4.1% on pass@k depending on the learning rate.

These experiments showcase that UpSkill even without correctness labels can offer strong improvements in pass@k that also generalizes OOD for some models.

To clarify the interpretability and intended usage of UpSkill, we would first like to emphasize that as we continue scaling UpSkill to a broader set of problems, we may not continue to find that all UpSkill skills are easily interpretable. Nevertheless, we expect skills to be interpretable in specific domains, and accordingly, we expanded our analysis of skill behavior and interpretability in the GSM8K setting, finding clear and distinguishable behaviors in each of the different skills such as overly detailed responses, narrative storytelling, and reasoning in code format. As each of these behaviors are targeted and individually may only perform well on a subset of problems, we do not recommend using UpSkill for pass@1 objectives. However, if needed, it is straightforward to generate a random completion for pass@1 testing by first sampling a random skill and then generating a completion with that skill.

Finally, we have further pronounced the distinction between our work and direct pass@k optimization (e.g., Chen et al.) by highlighting that UpSkill does not require correctness labels and encourages reproducible strategy diversity, and that these two approaches are likely orthogonal, as suggested by Chen et al.

We are grateful for the feedback we have received, and we believe these additions and revisions strengthen the paper and address all major reviewer concerns.



Thank you,

Authors

---

### Meta-Review · Area_Chair_urXq · 2026-01-04

**Summary:**

This paper propose to adopt the skill learning methods from previous reinforcement learning algorithms (e.g., DIAYN). A latent variable $z$ is learned using information theory-based objective functions to represent the "strategy" that steers the token distribution toward distinct modes. Experiments on GSM8K with GRPO show that UpSkill improves multi-attempt metrics for language models.

The work received diverse review opinions. While I appreciate the the paper for presenting an interesting theoretical idea (using mutual information for diversity), but the empirical evidence shows it is too brittle for publication in its current form. The method needs significantly more development to ensure its consistency and explainable performance gain.

**Reviewer Concerns:**

### Addressed Concerns

- Limited evaluation/scalability: The authors added experiments on an out-of-distribution (OOD) coding dataset(HumanEval) to address concerns about the method being confined to GSM8K.

- Stability & Llama performance: Concerns about negative results for Llama 3.1-8B were addressed by fixing a parsing bug in the evaluation code and introducing a KL penalty. New results show improved pass@k for Llama on GSM8K.

- Theoretical adherence: The authors introduced a negative KL estimator reward to better align the method with its theoretical assumptions.

- Comparison to direct pass@k optimization: The authors clarified that UpSkill is orthogonal to methods like Chen et al. as it does not require correctness labels and explicitly encourages strategy diversity.

- Computational overhead: The authors explicitly quantified the cost, stating it requires $O(k)$ forward passes.

### Remained concerns

- O.O.D. generalization for stronger models: While qwen improved on HumanEval, Llama 3.1-8B performance degraded on this O.O.D. task. This leaves the concern about the method's transferability to all strong base models partially unresolved.

- Effectiveness on distilled/small models: The authors noted that the R1-Distilled model performance decreased with UpSkill, indicating a potential limitation for smaller or highly optimized models.

**Reviewer Scores:**

- Reviewer Vey4 is likely to maintain the score since his/her confidence is low and the score is already acceptance.
- Reviewers CCs5 and ZGnj are also likely to maintain the recommendations since the consistency and interepretability issues of empirical results remain.

---

### Decision · Program_Chairs · 2026-01-26

Reject